# The influence of geostrophic strain on oceanic ageostrophic motion and surface chlorophyll

Zhengguang Zhang [1,2], Bo Qiu[3], Patrice Klein[4] & Seth Travis[3]

Oceanic submesoscale ageostrophic processes have been progressively recognized as an important upwelling mechanism to close the nutrient budget and sustain the observed primary production of phytoplankton in the euphotic layer. Their relatively small spatio-temporal scales (of 1~10 km and a few days) have hindered a systematic observational quantification of the submesoscale ageostrophic flow variability and its impact on ocean biogeochemistry. By combining surface drifters, satellite altimetry and satellite ocean-color data, we detect that when the strain rate of mesoscale surface geostrophic flow is strong, it favors a higher ageostrophic kinetic energy level and an increase in surface chlorophyll concentration. The strain-induced frontal processes are characterized by a surface chlorophyll increase and secondary ageostrophic upwelling along the light side of the oceanic density front. Further analysis indicates that the balanced ageostrophic motions with longer time scales are more effective in inducing chlorophyll increase than the unbalanced shorter time-scale wave motions.

[1] Physical Oceanography Lab, Qingdao Collaborative Innovation Center of Marine Science and Technology, Ocean University of China, Qingdao, China. [2] Laboratory for Ocean and Climate Dynamics, Qingdao National Laboratory for Marine Science and Technology, Qingdao, China. [3] Department of Oceanography, University of Hawaii at Manoa, Honolulu, HI, USA. [4] Laboratoire d'Océanographie Physique et Spatiale, Brest 29200, France. Correspondence and requests for materials should be addressed to B.Q. (email: bo@soest.hawaii.edu)

More than half of the primary production on Earth occurs in the surface layer of the ocean and involves photosynthetic fixation of carbon by phytoplankton[1]. Oceanic primary production is of fundamental importance because it sets a first-order constraint on the energy available to sustain oceanic ecosystems, and also provides a mechanism to remove carbon from the surface ocean by fixation and subsequent sinking of organic particles, playing a key role in oceanic uptake of atmospheric carbon dioxide. Biogeochemical estimates of new production surpass the apparent rate of nutrient supply by vertical mixing by a factor of 2 or more[2–4]. Oceanic mesoscale eddies with horizontal scale of tens to hundreds of kilometers are strongly constrained by the rotation of the Earth and are in geostrophic balance on the lowest order. In recent decades, a large number of studies aiming to identify the missing nutrients indicate that while the upwelling by oceanic mesoscale eddies could be an important contributor, its estimated contribution still accounts for only 20–30% of the annual requirement[3–11]. This discrepancy has stimulated the ongoing debate about what missed physical mechanisms could close the nutrient budget through additional vertical nutrient supply.

One critical candidate towards closing the nutrient budget is the oceanic submesoscale ageostrophic motion that can emerge from the straining field of interacting mesoscale eddies. Submesoscale processes are particularly relevant to phytoplankton productivity because the time-scales on which they act are similar to those of phytoplankton growth[12,13]. Their dynamics are associated with motions occurring on spatial scales of 1–10 km and temporal scales of a few inertial periods (~days). They can break down the geostrophic balance that suppresses vertical motions[14], and support vertical velocities as large as 10–100 m day$^{-1}$, an order of magnitude larger than the vertical velocities induced by mesoscale eddies[15–18]. Thus, submesoscale processes can play a crucial role in transporting nutrients into the sunlit ocean for phytoplankton production[13]. The vertical nutrient fluxes by submesoscale processes can be as large as the contribution by mesoscale eddies based on a global theoretical estimation[19]. In addition, high-resolution coupled physical–biogeochemical models have shown primary production to increase by up to a factor of 3 when submesoscale features are resolved[8,20].

The submesoscale ageostrophic motions are associated with a wide range of dynamical processes: these include unbalanced wave motions, such as near inertia waves and internal gravity waves, and balanced non-wave motions, such as the ageostrophic submesoscale vortices and filaments emerging from frontogenetic instability[21,22] and mixed-layer instability[23]. Although the vertical velocity of the wave motions can be quite large, it is often too fast evolving and cannot provide the sustained nutrient upwelling necessary for uptake by the near-surface phytoplankton. In contrast, the balanced submesoscale frontal processes involve the release of the potential energy of mesoscale fronts around eddies and can sustain vertical secondary circulation with time-scales comparable with the nutrient uptake by phytoplankton[13]. The mesoscale deformation flow, usually referred to as the geostrophic strain field, can thus effectively enhance the mesoscale fronts, dominate the development of submesoscale frontal ageostrophic perturbations, and play an important role in controlling nutrient upwelling and oceanic primary production.

Despite the crucial importance of oceanic submesoscale processes, our understanding of the submesoscale processes and quantification of their biogeochemical impacts remain fragmentary. The relatively short spatio-temporal scales of submesoscale processes prevent synoptic observations of them either by moored arrays or ship surveys. At the same time, both the spatial resolution of along track satellite observations and the 100–300 km spacing between the altimeter satellite ground tracks miss the submesoscale footprint. Difficulties associated with the simultaneous measurement of submesoscale features can be circumvented through the use of Lagrangian-based observations. Nowadays, the number of global positioning system (GPS)-tracked surface drifters is large enough to achieve a global coverage and can provide accurate real-time position time series[24–28]. With a typical sampling interval at about 6 h, the drifters are able to resolve the submesoscale ageostrophic velocity signals at the sea surface globally; this includes both the unbalanced wave motions[29] and submesoscale balanced ageostrophic motions[27,30].

In addition, the Lagrangian-based observations can in addition benefit the quantification of the biogeochemical responses. An important method to observe phytoplankton distribution at a global scale is through satellite ocean-color remote sensing. From a Eulerian approach, the phytoplankton variability is dominated by horizontal advection and stirring rather than biological processes[31–34]. Thus, a major challenge when using satellite ocean-color data to study phytoplankton dynamics is to untangle the footprint of biological processes (such as primary production) from that of physical processes (such as advection and stirring). By following water particle trajectories, Lagrangian analysis provides a new perspective that naturally takes into account the effect of advection[35]. For example, by utilizing the virtual trajectories constrained by the altimetry geostrophic velocity field, efforts have been made to quantify the variations in phytoplankton productivity over spatio-temporal scales of days and tens of kilometers[36]. However, these efforts are limited by the insufficient resolution of the current altimeter data products, and by the necessity of making the geostrophic approximation without taking into account the ageostrophic velocity components[35]. With the use of GPS-tracked surface drifters to provide trajectories of water particles and the combined satellite altimeter and ocean-color data, we have now a unique opportunity to explore the connection between the oceanic fronts deformed by the geostrophic strain field and the submesoscale ageostrophic processes along with the associated biogeochemical response based on the observational data alone.

Here we show that there exists a positive relationship between the ageostrophic kinetic energy and the near-surface chlorophyll variation depending on the local geostrophic strain field. By adopting a composite analysis with a global coverage, we clarify the spatial structure of the ageostrophic motion and the chlorophyll response around the mesoscale front under strong strain deformation, which are characterized by a surface chlorophyll increase and secondary ageostrophic upwelling along the light side of the oceanic density front. Finally, our analysis indicates that the balanced ageostrophic motions with longer time-scales are more effective in inducing chlorophyll increase than the unbalanced shorter time-scale wave motions.

## Results

**Geostrophic strain and chlorophyll variation.** Mesoscale eddies with horizontal scales of tens to hundreds of kilometers are well known to be the dominant reservoir of kinetic energy of the world ocean[37]. They serve as a principal sink for the energy of planetary-scale mean oceanic circulation through balanced instabilities, for example, the quasi-geostrophic barotropic and baroclinic instabilities[38]. The geostrophic kinetic energy of mesoscale eddies is relatively strong in regions where the energetic larger-scale currents are unstable, for example, along the Western Boundary Current systems (e.g., Kuroshio Extension and the Gulf Stream), the Subtropical Countercurrent and the Antarctic Circumpolar Current as shown by the global distribution of geostrophic kinetic energy $E_g$ in Fig. 1a. These regions are

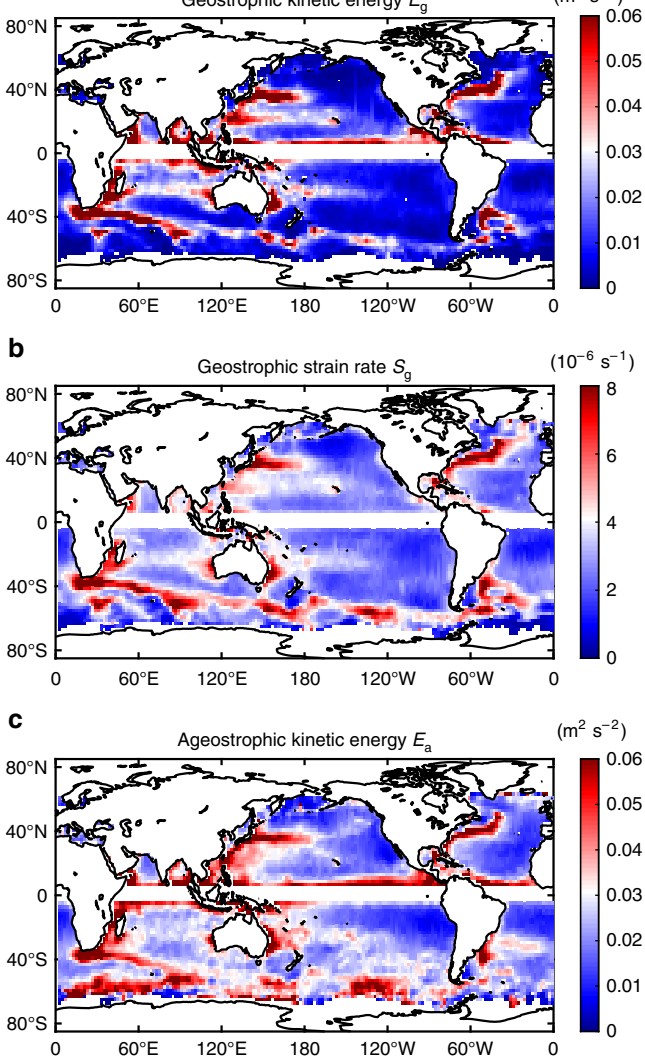

**Fig. 1** Global distributions of mean energies and strain rate. **a** Geostrophic kinetic energy $E_g$. **b** Geostrophic strain rate $S_g$. **c** Ageostrophic kinetic energy $E_a$. All maps are constructed by averaging with 3° × 3° moving window at each grid point for all available data points. Notice that the geostrophic kinetic energy $E_g$ and strain $S_g$ are computed by using the altimeter data, whereas the ageostrophic kinetic energy $E_a$ is derived from the drifter-observed absolute surface velocity data in combination with the altimeter data (see Methods). Source data are provided as a Source Data file

characterized by energetic background currents and abundant mesoscale eddies, and the regional eddy-eddy and eddy-mean flow interactions are expected to be similarly vigorous. The resultant strong geostrophic deformation fields are demonstrated by hotspots of geostrophic strain rate $S_g$ within these regions, as shown in Fig. 1b. The detailed definition for $E_g$ and $S_g$ can be found in the Methods section.

The strain flow field is characterized by a local saddle point of geostrophic stream function, acting to stretch flow in one direction and compress it in the perpendicular direction. It works to continuously shrink the spatial scale of the horizontal front and enhance the horizontal gradient (e.g., density gradient) along the compressed direction. As the scale shrinks, a point is reached when the geostrophic balance can no longer be held, and ageostrophic motions emerge, cascading the mesoscale energy to

smaller scales[12,18]. Within the regions with abundant mesoscale eddies and vigorous geostrophic strain, the energy sources and straining conditions all promote effective development of ageostrophic perturbations, leading to the enhancement of the ageostrophic kinetic energy level $E_a$ as shown in Fig. 1c. (Detailed definition for $E_a$ can be found in the Methods section.) It should be noted that the ground tracks of currently operating altimeter satellites are unable to resolve relatively small-scale geostrophic signals and potentially introduce errors in the ageostrophic velocity estimation. It has, however, been shown that this does not substantially bias the results when investigating the ageostrophic motions and their relation to geostrophic strain rates[30].

In addition to providing information for the ageostrophic motions as shown in Fig. 1c, the surface drifters also provide Lagrangian trajectories of ocean surface water particles. When combined with the ocean-color remote sensing, drifters allow a Lagrangian investigation of the near-surface chlorophyll variation. As mentioned before, the nature of Lagrangian observation has already taken the effect of horizontal advection and stirring into account, allowing us to focus on the effect of vertical processes and biological production taking place along tracked water particles. Considering that chlorophyll concentrations can vary by several orders of magnitude, the satellite observed chlorophyll concentration Chl (unit: mg m$^{-3}$) value is expressed below by its base 10 logarithm log Chl. After projecting the satellite-observed chlorophyll concentration onto the drifter trajectories, the Lagrangian chlorophyll variation rate D(log Chl)/D$t$ can be readily estimated. The original drifter data is used rather than the gridded data. If D(log Chl)/D$t$ > 0, the chlorophyll increases within the water particle tracked by the drifter (see Methods).

After calculating the Lagrangian chlorophyll variation rate and the geostrophic/ageostrophic kinetic energy information along the drifter trajectories, their relations can be investigated. The globally averaged curve of D(log Chl)/D$t$ changing with ageostrophic energy $E_a$ exhibits a clear increasing tendency as shown in Fig. 2a. The chlorophyll variation rate D(log Chl)/D$t$ is positive when the ageostrophic energy $E_a$ > 0.1 m$^2$ s$^{-2}$, indicating that a strong ageostrophic kinetic energy favors an increase in local chlorophyll. When the ageostrophic kinetic energy is strong, a typical rate D(log Chl)/D$t$ ~O(10$^{-2}$ day$^{-1}$) indicates that the chlorophyll concentration Chl can increase by 10-fold in 100 days. Although the ageostrophic kinetic energy $E_a$ ranges from 0 to 1 m$^2$ s$^{-2}$ in Fig. 2a, about 18% of the data points are found to have $E_a$ >0.1 m$^2$ s$^{-2}$. This means only a small portion of the high ageostrophic events can effectively contribute to the chlorophyll increasing. The percentage of the data with $E_a$ >0.1 m$^2$ s$^{-2}$ can reach 40–50% within the strong-current regions; and in the subtropical gyres, about 20–30% data points have large enough ageostrophic energy to lead to a chlorophyll increase (Supplementary Note 1). When compared with the ageostrophic kinetic energy, the globally averaged curve of D(log Chl)/D$t$ as changing with the geostrophic energy $E_g$ exhibits no significant increasing tendency above the error bar as shown in Fig. 2b. This does not mean that mesoscale eddies are unimportant for the chlorophyll increase. Rather, it means that the Lagrangian chlorophyll variation rate D(log Chl)/D$t$ are more sensitive to the local ageostrophic energy rather than the geostrophic energy. For example, in some regions, mesoscale eddies have the strongest chlorophyll response located at the center of the eddy, where the geostrophic kinetic energy is relatively low[39].

Rather than the geostrophic kinetic energy, we find the geostrophic strain rate to be more relevant to the biogeochemical responses. The globally averaged curves of D(log Chl)/D$t$ changing with the geostrophic strain rate $S_g$ exhibit a clear

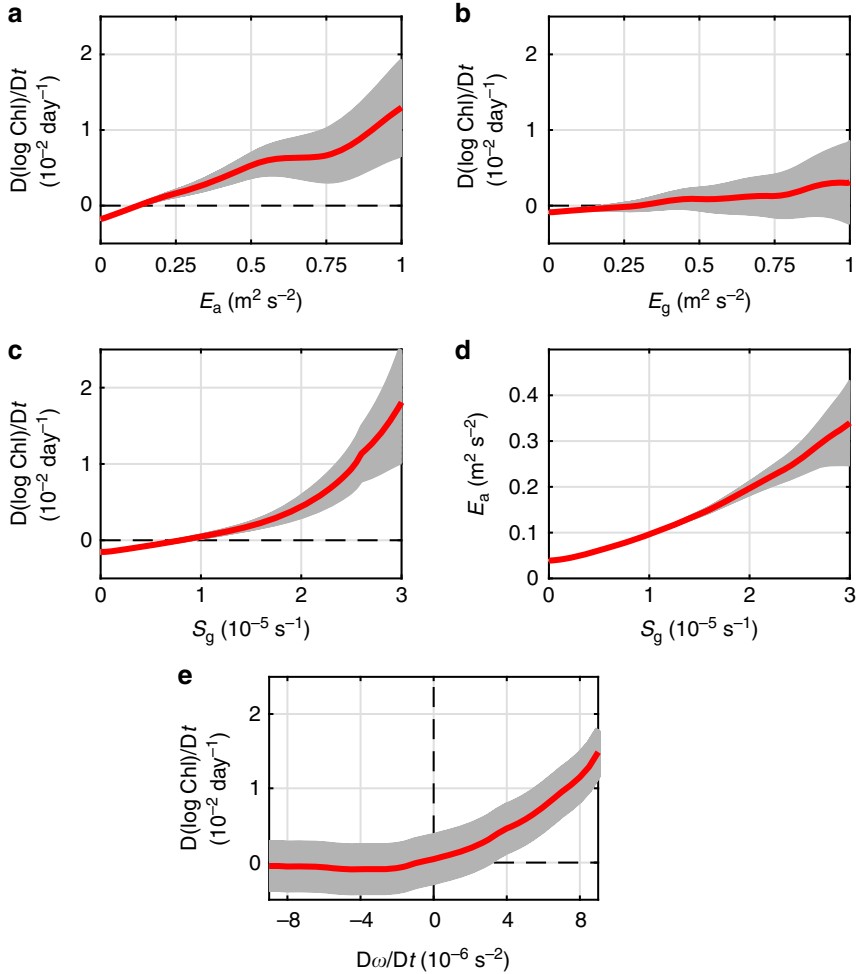

**Fig. 2** Chlorophyll variation rate vs. kinetic energy and strain rate. Globally averaged curves of Lagrangian chlorophyll variation rate D(log Chl)/D$t$ as a function of local **a** ageostrophic kinetic energy $E_a$, **b** geostrophic kinetic energy $E_g$, and **c** geostrophic strain rate $S_g$. **d** Globally averaged curve of ageostrophic kinetic energy $E_a$ as a function of local geostrophic strain rate $S_g$. **e** Globally averaged curves of Lagrangian chlorophyll variation rate D(log Chl)/D$t$ as a function of the Lagrangian derivative of modified geostrophic vorticity D$\omega$/D$t$. In **a**, **b**, D(log Chl)/D$t$ is composited against $E_a$ and $E_g$ using a moving average window with a width 0.05 m$^2$ s$^{-2}$. In **c**, **d**, $E_a$ and D(log Chl)/D$t$ are composited against $S_g$ using a moving average window with a width 0.1 × 10$^{-5}$ s$^{-2}$. In **e**, D(log Chl)/D$t$ are composited against the D$\omega$/D$t$ using a moving average window with a width 0.5 × 10$^{-6}$ s$^{-2}$. The red curve in each subfigure represents the average curve and the light-gray shading represents the error bar computed by the standard error of average as Std/$N^{1/2}$, where Std and $N$ are the standard deviation and data number within each averaging bin, respectively. Source data are provided as a Source Data file

increasing tendency as shown in Fig. 2c. The chlorophyll variation rate D(log Chl)/D$t$ is positive when the geostrophic strain rate $S_g > 1.0 \times 10^{-5}$ s$^{-1}$, indicating that a strong geostrophic strain rate, like the strong ageostrophic kinetic energy, also favors an increase of local chlorophyll. The amplitude of D(log Chl)/D$t$ can also reach O(10$^{-2}$ day$^{-1}$), comparable to the chlorophyll variation rate in association with strong ageostrophic energy. This result is consistent with the expectation that the geostrophic strain shrinks frontal scales and promotes development of ageostrophic perturbations. As shown in Fig. 2d, the increasing curve of the ageostrophic energy $E_a$ changing with the geostrophic strain rate $S_g$ confirms this expectation, suggesting that the mesoscale strain field could be a key player in regulating the ageostrophic energy and resulting in increasing chlorophyll concentration. Recent high-resolution OGCM simulations have shown that the ageostrophic energy and associated vertical transport undergo substantial seasonal variation[40], and similar seasonality is observed for the relation between the chlorophyll variation rate and geostrophic strain rate defined here (Supplementary Note 2).

Besides the strain-induced ageostrophic motions causing a chlorophyll increase, mesoscale divergence could also induce a surface chlorophyll response[41,42]: when a cyclonic eddy enhances or an anticyclonic eddy decays, the isopycnal surfaces are expected to be uplifted, resulting in a surface chlorophyll increase. In contrast, when an anticyclonic eddy enhances or a cyclonic eddy decays, downward motion of isopycnal surfaces is expected, which can result in a surface chlorophyll decrease. The mesoscale divergence is closely related to the Lagrangian variation of the relative vorticity. By defining a modified geostrophic relative vorticity $\omega$ and its Lagrangian derivative D$\omega$/D$t$ in the Methods sections, D$\omega$/D$t$ > 0 is always associated with mesoscale upwelling in the northern and southern hemispheres (D$\omega$/D$t$ < 0 for downwelling). As shown in Fig. 2e, the chlorophyll variation rate D(log Chl)/D$t$ is positive when the Lagrangian derivative of $\omega$ is positive. The chlorophyll variation rate D(log Chl)/D$t$ is zero or even negative when the Lagrangian derivative of $\omega$ is negative. This result is consistent with the physical expectations based on former studies regarding the relationship between enhancing/weakening

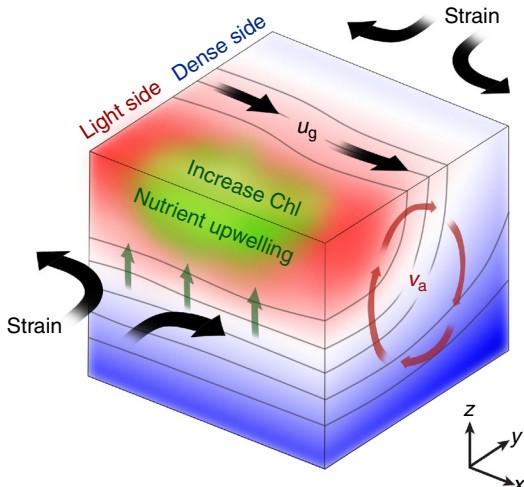

**Fig. 3** Strain-induced frontal processes in the Northern Hemisphere. A mesoscale front exists along the $x$-direction with dense (light) water present in positive (negative) $y$ direction. In the Northern Hemisphere, the along-front geostrophic velocity $u_g$ is pointed to the positive direction of $x$. Due to the background geostrophic strain field stretching along the front and compressing across the front, the cross-front density gradient is enhanced by the strain field. As the cross-front density gradient enhances, frontal instabilities develop causing release of available potential energy of the front and restratification. The resulting net effect sustains a cross-front ageostrophic secondary circulation in the $y$–$z$ plane as indicated by $v_a$. The secondary circulation causes upwelling/downwelling along the light/dense side of the front, generating upward nutrient fluxes and a chlorophyll increase along the light side of the front

mesoscale eddies and upwelling/downwelling[41,42]. This result also confirms the validity of the chlorophyll variation rate D(log Chl)/D$t$ and allows us to focus on the detailed structure of the strain-induced ageostrophic motions and their surface chlorophyll response.

**Strain-induced frontal processes**. The most effective way for strain to strengthen a geostrophic front and stimulate ageostrophic perturbations is to stretch in the along-front direction and compress in the cross-front direction, as sketched in Fig. 3. During this process, the background strain field continuously enhances the cross-front density gradient. As the cross-front density gradient strengthens and the cross-front scale shrinks, frontal instabilities develop through the release of the available potential energy of the front. The frontal instabilities generate an ageostrophic secondary circulation that has an upwelling/downwelling along the light/dense side of the front and a cross-front surface ageostrophic velocity from light side to dense side. This cross-front secondary circulation is well documented by frontal instability theories[21–23,43], diagnostic vertical velocity estimation[18,44–46], and high-resolution numerical simulations[15–17,47]. The upward nutrient flux and the resultant chlorophyll increase along the light side of the front are also expected[12,47–49].

The importance of strong geostrophic strain for chlorophyll increase is highlighted in the previous section. In order to clarify the connections among the frontal processes, the ageostrophic motion and the chlorophyll variation based on the observational data, we search for strain saddle points as the local maximum points of geostrophic strain rate with $S_{gc} > 1.0 \times 10^{-5}\,\text{s}^{-1}$ and saddle points for the geostrophic stream function from the altimetry data. In order to strengthen the geostrophic front and

promote ageostrophic perturbations, the stretching direction of the local strain field needs be aligned with the front. Thus, only the strain saddle points with their stretching direction closely aligned with the local geostrophic velocity direction will be taken into account (see Method for the specific criterion). The composite analysis is used to investigate the spatial structure around the strain saddle points. Before the composition, all properties are normalized by the centered strain rate $S_{gc}$ and the coordinates are rotated such that the along-front geostrophic velocity points to the positive $x$-direction. Since the direction of the frontal ageostrophic secondary circulation depends on the sign of the local Coriolis parameter, the Northern and Southern Hemispheres are composited separately.

The composited properties in the rotated along-front coordinate ($x_r,y_r$) of the northern hemisphere are shown in Fig. 4. As shown in Fig. 4a, the composited normalized strain rate $S_{gn}$ reaches its maximum with a unit amplitude at the center point of the rotated coordinate. Interestingly, there exists a positive tail of $S_{gn}$ along the front in the downstream direction. The normalized geostrophic velocity field ($u_{gn},v_{gn}$) in Fig. 4b exhibits the strongest geostrophic velocity along the front, consistent with the definition of the along-front coordinate. There is also an identifiable geostrophic strain field that stretches in the along-front direction and compresses in the cross-front direction. The normalized sea surface temperature anomaly (SSTA) $T_{an}$ in Fig. 4c demonstrates that the light part of the front is on right-hand side of the along-front current and the dense part on the left-hand side, consistent with the northern hemisphere geostrophic balance. For estimation of the magnitude of the results, we take the amplitude of strain rate to be $O(10^{-5}\,\text{s}^{-1})$, giving a rough amplitude of the geostrophic velocity of about $0.5\,\text{m}\,\text{s}^{-1}$ and the cross-front temperature difference of about 2 °C.

The normalized ageostrophic kinetic energy $E_{an}$ exhibits a monopolar positive center structure around the strain saddle point as shown in Fig. 4d. This is consistent with the result in Fig. 2d that the stronger geostrophic strain rate favors a higher ageostrophic kinetic energy level. The normalized Lagrangian chlorophyll variation rate D(log Chl$_n$)/D$t$ shown in Fig. 4e exhibits a dipolar structure: the region of strongly increasing chlorophyll is located on the right-handed light side of the front, with a weaker decreasing region located on the left-handed dense side of the front. This is consistent with the dynamical expectation, as illustrated in Fig. 3, that there is upwelling along the light side of the front, bringing subsurface nutrients to the ocean surface and increasing the surface chlorophyll. The vertical extension of this ageostrophic upwelling is mostly confined to the local mixed-layer depth. Since the maximum of chlorophyll concentration is located in subsurface layers[49], there is a possibility that this observed chlorophyll increase could also be caused by upwelling of subsurface waters carrying phytoplankton with higher chlorophyll concentration, and is not indicative of actual new primary production. Although the two effects cannot be separated with only the current observational data, the observed chlorophyll increase is always aligned with the ageostrophic upwelling.

According to Fig. 3, there will be a surface branch of the ageostrophic secondary circulation in the cross-front direction, which flows from the light side to the heavy side at surface. As shown by normalized cross-front ageostrophic velocity $v_{an}$ in Fig. 4f, there is a positive cross-front ageostrophic velocity center located around the strain saddle point, in agreement with the theoretical expectations. Its amplitude is about one order of magnitude smaller than the along-front geostrophic velocity. Additionally, there is also a negative center of $v_{an}$ in Fig. 4f on the downstream side of the strain saddle point, which can

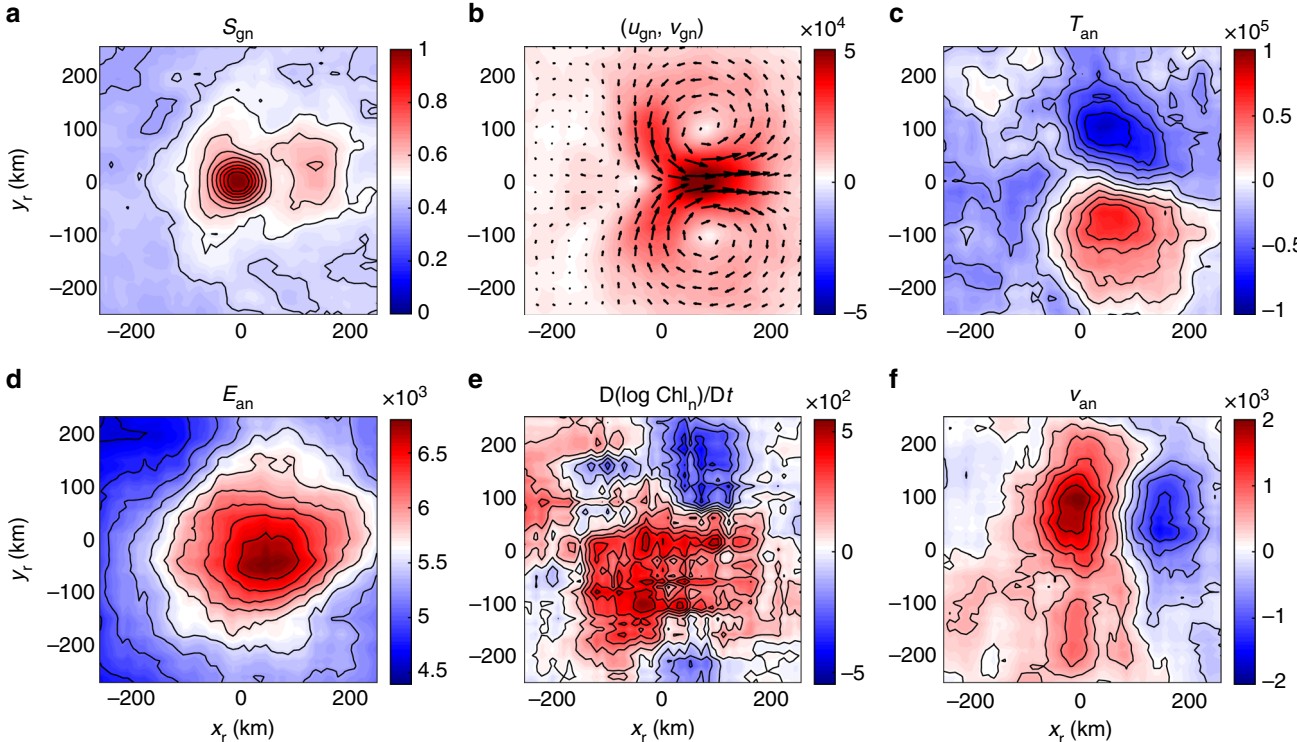

**Fig. 4** Northern Hemisphere frontal processes and strong geostrophic strain. Composited distributions of normalized properties in the along-front coordinate: **a** normalized geostrophic strain rate $S_{gn}$, **b** normalized geostrophic velocity field $(u_{gn}, v_{gn})$, where the color represent the speed of the geostrophic flow, and the vectors represent the geostrophic velocity field, **c** normalized surface temperature anomaly $T_{an}$, **d** normalized ageostrophic kinetic energy $E_{an}$, **e** normalized Lagrangian chlorophyll variation rate $D(\log Chl_n)/Dt$, and **f** normalized cross-front ageostrophic velocity $v_{an}$. All properties are normalized by the strain rate at the centered strain saddle point $S_{gc}$. Thus, $S_{gn}$ and $D(\log Chl_n)/Dt$ are dimensionless, and the units of $(u_{gn}, v_{gn})$, $T_{an}$, $E_{an}$, and $v_{an}$ are $m^1$, °C $s^{-1}$, $m^2$ $s^{-1}$, and $m^1$, respectively. The along-front and cross-front directions are $x_r$ and $y_r$, respectively. All parameters are composited using a moving window with a bin size 25 km × 25 km. Source data are provided as a Source Data file

emerge from the frontolysis processes caused by the cross-front stretching (along-front squeezing) at this location. If we take the amplitude of strain rate to be $O(10^{-5} s^{-1})$, the rough amplitude of the Lagrangian chlorophyll variation rate $D(\log Chl)/Dt$ is $O(10^{-2} day^{-1})$ and the cross-front ageostrophic velocity is about 2 cm s$^{-1}$. If we further take the width of the front to be 20 km, the cross-front ageostrophic velocity was found to be 2 cm s$^{-1}$ and the vertical scale of the secondary circulation to be 50 m, a typical mixed-layer depth and relevant vertical scale of motion, then the resulting vertical velocity will be about $2 \times 10^{-4}$ m s$^{-1}$ or 20 m day$^{-1}$, which is consistent with former estimations of vertical velocity in mesoscale stirring region[50].

The composite results for the Southern Hemisphere are shown in Fig. 5. The distributions of the geostrophic strain rate, the geostrophic velocity, and the ageostrophic kinetic energy are almost the same as those in the Northern Hemisphere. Since the Coriolis parameter is negative in the Southern Hemisphere, the light side of the front is located to the left of the along-front current as shown in the Fig. 5c. Because the ageostrophic secondary circulation always works to flatten the density surface and release the potential energy of the front, its surface branch is always from the light side of the front to the dense side as shown by the negative cross-front ageostrophic velocity in Fig. 5f. The corresponding ageostrophic upwelling is expected along the light side of the front, which is consistent with the composited chlorophyll variation rate pattern in Fig. 5e: the dipole structure of the chlorophyll variation rate exhibits an increasing center on the left-hand side of the along-front current. The amplitude of the

composited properties in the Southern Hemisphere is about the same as those in the Northern Hemisphere.

## Discussion
By combining the surface drifter and satellite remote-sensing data, we show that the Lagrangian chlorophyll variation rate depends positively on the local ageostrophic kinetic energy level and the geostrophic strain rate, as the strong strain rate enhances the local ageostrophic kinetic energy and favors an increase in near-surface chlorophyll. Further composite analysis reveals that the spatial structure of the strain-induced frontal processes is characterized by a cross-front ageostrophic secondary circulation with upwelling and a chlorophyll increase along the light side of the density front. Such an ageostrophic secondary circulation is basically a balanced motion rather than unbalanced wave motions[51,52].

Vertical velocities associated with the wave motions are often too fast evolving and cannot provide sustained nutrients upwelling for complete uptake by near-surface phytoplankton. In contrast, the balanced submesoscale frontal processes can maintain vertical secondary circulation lasting longer than several inertial periods, allowing for more complete nutrient uptake by the near-surface phytoplankton. Therefore, it can be expected that the chlorophyll variation rate depends on time-scales of the ageostrophic motions. We can use a filter to separate the ageostrophic velocity observed by the drifters into high-pass and low-pass filtered components. Our selection of the cut-off period of the filter follows two criteria: first, the cut-off period should be

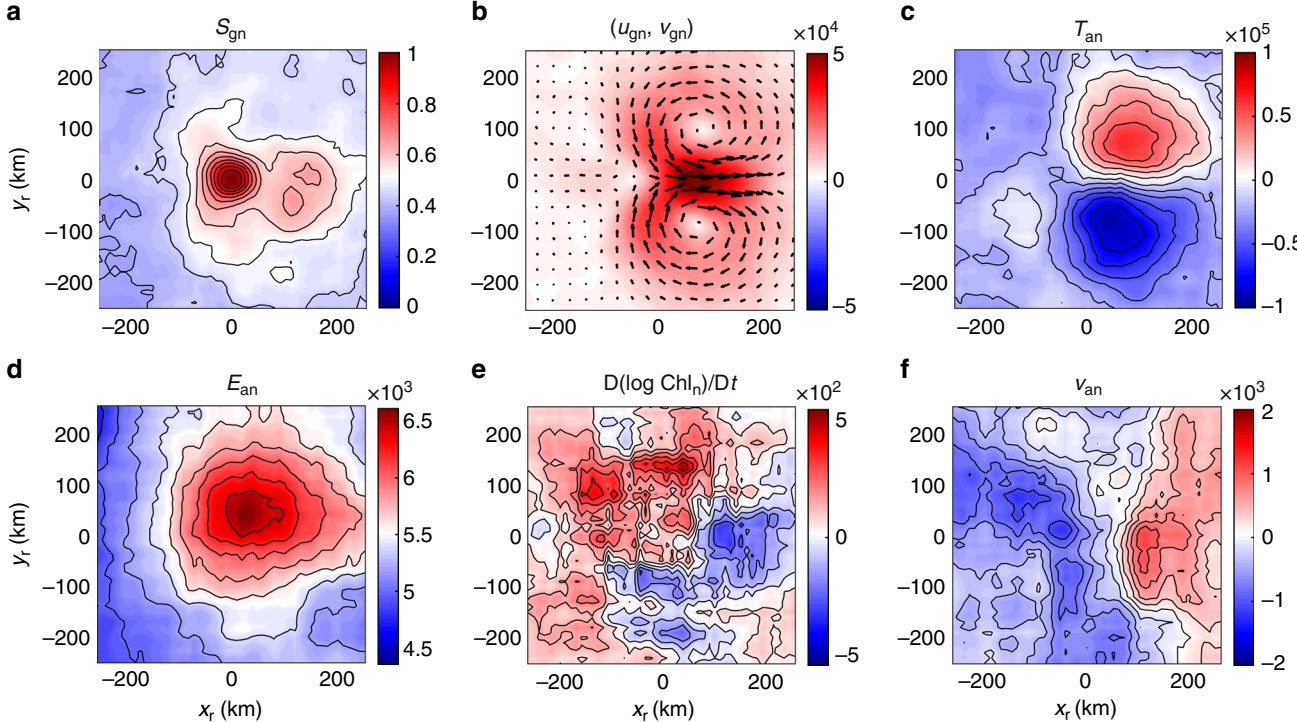

**Fig. 5** Southern Hemisphere frontal processes and strong geostrophic strain. Composited distributions of normalized properties in the along-front coordinate: **a** normalized geostrophic strain rate $S_{gn}$, **b** normalized geostrophic velocity field $(u_{gn}, v_{gn})$, where the color represent the speed of the geostrophic flow, and the vectors represent the geostrophic velocity field, **c** normalized surface temperature anomaly $T_{an}$, **d** normalized ageostrophic kinetic energy $E_{an}$, **e** normalized Lagrangian chlorophyll variation rate $D(\log Chl_n)/Dt$, and **f** normalized cross-front ageostrophic velocity $v_{an}$. All properties are normalized by the strain rate at the centered strain saddle point $S_{gc}$. Thus, $S_{gn}$ and $D(\log Chl_n)/Dt$ are dimensionless, and the units of $(u_{gn}, v_{gn})$, $T_{an}$, $E_{an}$, and $v_{an}$ are $m^1$, $°C\ s^1$, $m^2\ s^{-1}$, and $m^1$, respectively. The along-front and cross-front directions are $x_r$ and $y_r$, respectively. All parameters are composited using a moving window with a bin size $25\ km \times 25\ km$. Source data are provided as a Source Data file

longer than the typical inertial period, which is the upper bound for the period of internal gravity waves, near inertial waves and major tidal motions. Second, the cut-off period should give equal energy partition between both the high-pass and low-pass filtered ageostrophic motions, allowing us to compare the chlorophyll responses for the high-pass/low-pass filtered ageostrophic motions under a uniform standard. Based on this second criterion, the high-pass/low-pass filtered ageostrophic motions reach an equal energy level when the cut-off period is 7 days (Fig. 6a). Considering that the 7-day period is longer than the typical inertial period, it serves as a reasonable choice of the cut-off period. Using this cut-off period, we compute the Lagrangian chlorophyll variation rates as a function of high-pass and low-pass filtered ageostrophic energy level, respectively. As shown in Fig. 6b, c, the chlorophyll response for the low-pass ageostrophic energy is much larger than the high-pass ageostrophic energy, in agreement with expectations. The positive Lagrangian chlorophyll variation rate $D(\log Chl)/Dt$ of the low-pass ageostrophic energy can even reach the amplitude of $3 \times 10^{-2}$ day$^{-1}$. This means that the chlorophyll concentration Chl can increase by 10-fold in 1 month when the low-pass ageostrophic kinetic energy is particularly strong. Because the low-pass filtered velocities mostly contain the balanced part of the ageostrophic motions, this result suggests that the balanced ageostrophic motions with longer time-scales are more effective in bringing about chlorophyll increase and possible net phytoplankton growth.

Since the chlorophyll increase is relevant to the primary production, we define the Lagrangian chlorophyll-increasing rate

as the positive Lagrangian chlorophyll variation rate $D(\log Chl)/Dt$ (>0). Considering that the Lagrangian chlorophyll variation rate increases at the higher end of both the high-pass/low-pass ageostrophic energy levels, we can compute averaged chlorophyll-increasing rates for strong/weak ageostrophic energy level as $D(\log Chl_{strong})/Dt$ and $D(\log Chl_{weak})/Dt$, respectively (see Methods about the separation method). The difference between these two increasing rates $D(\log Chl_{diff})/Dt$ gives an indicator of how the chlorophyll-increasing rate changes with the ageostrophic energy level. The global distribution of this indicator $D(\log Chl_{diff})/Dt$ is computed for both the high-pass and low-pass ageostrophic energy values, as shown in Fig. 7a, b. Again, the low-pass ageostrophic energy exhibits stronger chlorophyll response, which is consistent with the results in Fig. 6. The global pattern of $D(\log Chl_{diff})/Dt$ shares similar hotspots as the global map of ageostrophic kinetic energy in Fig. 1c, suggesting the higher ageostrophic energy along major current systems and some significant topographic features favor higher chlorophyll-increasing rate and promote primary production.

The results of this study highlight the effect of geostrophic strain and submesoscale ageostrophic processes on the near-surface chlorophyll variations. The spatial structure of the strain-induced frontal ageostrophic secondary circulation and the chlorophyll response are obtained based on available observational data. Our results shed new light onto the problem of how to close the surface nutrient budget in order to sustain the observed level of oceanic primary production. Since the submesoscale ageostrophic motions are typically not included

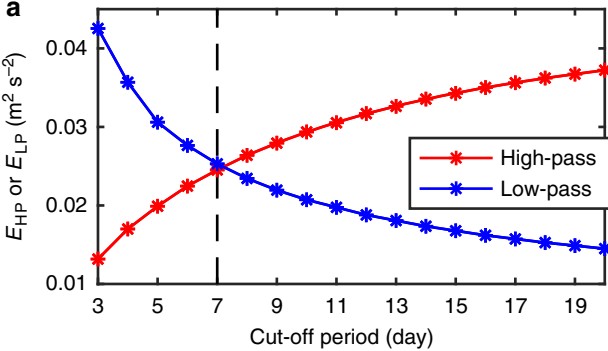

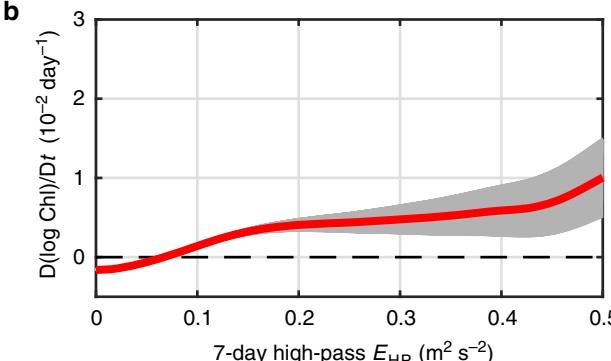

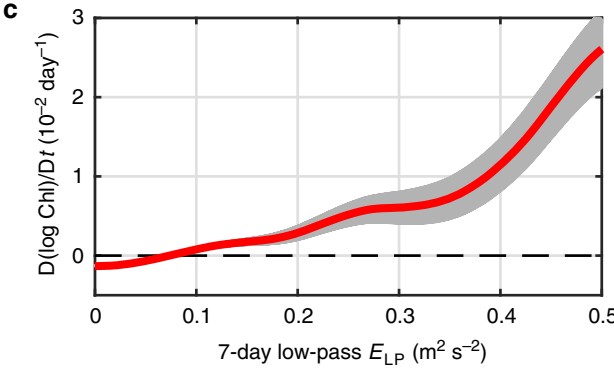

**Fig. 6** Chlorophyll variation rate changing with ageostrophic kinetic energy. **a** Globally averaged high-pass and low-pass filtered ageostrophic kinetic energy changing with the cut-off period of the filter. The dashed line at 7-day denotes the cut-off period where the high-pass/low-pass filtered ageostrophic kinetic energy levels are equal. **b** Globally averaged curves of Lagrangian chlorophyll variation rate D(log Chl)/Dt as a function of 7-day cut-off high-pass ageostrophic kinetic energy $E_{HP}$. **c** Same as **b**, but for 7-day cut-off low-pass ageostrophic kinetic energy $E_{LP}$. The chlorophyll variation rate is composited against the high-pass and low-pass ageostrophic kinetic energy $E_{HP}$ and $E_{LP}$, using a moving average window with a width 0.05 m² s⁻². The red curve in each subfigure represents the average curve and the light-gray shading represents the error bar computed by the standard error of the average. Source data are provided as a Source Data file

in the present-day global carbon cycle models, a key question arising naturally is whether their representation is important for the structuring of oceanic ecosystems, the uptake of atmospheric $CO_2$, and the large-scale distributions of properties in the ocean. Our results here may help to establish parameterizations of the effect of mesoscales–submesoscales or provide an observational base line to test the output of numerical models. Our investigation here has benefited from the combined use

of surface drifters and satellite remote sensing. We hope this study will motivate future studies to gain a better understanding of the oceanic submesoscale processes and the ocean ecosystems, when the next-generation Surface Water Ocean Topography satellite mission is underway, giving observations of submesoscale signals with a horizontal resolution down to ~15 km[53–55].

## Methods

**Altimetry dataset.** The SSALTO/DUACS delay-time altimetry product provided by AVISO is used here (issue 5.0 updated 2016/08/20). This multiple-satellite-merged data contains global gridded daily sea surface height, sea level anomaly, sea surface geostrophic velocity anomaly ($u_g$, $v_g$) with a ¼ degree resolution from year 1993 to 2016.

**Surface drifter data.** The drifter data used here is provided by the Drifter Data Assembly Center (DAC) of National Oceanic and Atmospheric Administration. The DAC assemble and provide uniform quality control of sea surface temperature (SST) and surface velocity by satellite-tracked surface drifting buoy observations. The surface velocity ($u$, $v$) and SST $T$ measurements are provided for every 6 h from year 1993 to 2011. They have a global coverage and contain a total of 22,249,337 observational data points. The SSTA is computed as $T_a = T - T_0$, where $T_0$ is the climatological monthly mean SST at the drifter location.

**Chlorophyll data.** The chlorophyll data of case-1 water is provided by the ESA GlobColour Project (version 4.1, updated 31 August 2017), which merges several sensors with ¼ degree spatial resolution of daily data[56] from year 1998 to 2017. The unit for chlorophyll concentration Chl is mg m⁻³ and all chlorophyll values in this study are expressed by their base 10 logarithm log Chl.

**Geostrophic kinetic energy and strain.** The surface geostrophic kinetic energy can be computed from altimeter-derived geostrophic velocity anomaly as $E_g(t) = (u_g^2 + v_g^2)/2$.

The geostrophic strain rate $S_g$ is computed by the surface geostrophic velocity anomaly ($u_g$, $v_g$) from the altimeter data:

$$S_g = \sqrt[2]{\left(\frac{\partial u_g}{\partial x} - \frac{\partial v_g}{\partial y}\right)^2 + \left(\frac{\partial v_g}{\partial x} + \frac{\partial u_g}{\partial y}\right)^2}. \quad (1)$$

The geostrophic strain flow field is characterized by stretching along one direction and squeezing along the perpendicular direction. The stretching direction is given by the principal axis of the strain rate tensor and the azimuth angle of the principal axis of the strain rate tensor is calculated by:

$$\theta_s = \frac{1}{2}\tan^{-1}\left(\frac{\partial u_g/\partial y + \partial v_g/\partial x}{\partial u_g/\partial x - \partial v_g/\partial y}\right). \quad (2)$$

Similarly, the azimuth angle of the local geostrophic velocity is given by:

$$\theta_g = \tan^{-1}\left(v_g/u_g\right). \quad (3)$$

**Ageostrophic velocity and kinetic energy.** The satellite-tracked drifters observe the absolute horizontal velocity $\bar{u}$ at ocean surface. In order to separate the signals of ageostrophic motions, the ocean surface ageostrophic velocity is computed as $\mathbf{u_a} = \mathbf{u} - \mathbf{u_g} - \mathbf{u_0}$, where $\mathbf{u_g}$ is the simultaneous altimetry geostrophic velocity anomaly at the drifter location and $\mathbf{u_0}$ is the climatological mean surface velocity computed from the drifter absolute velocity $\mathbf{u}$. The ageostrophic kinetic energy is computed as $E_a = \mathbf{u_a}^2/2$ along each drifter trajectory.

When a cut-off period is set, the high-pass filter can be applied to $\mathbf{u_a}$ to compute the high-pass filtered ageostrophic velocity $\mathbf{u_{HP}}$. The residual velocity $\mathbf{u_{LP}} = \mathbf{u_a} - \mathbf{u_{HP}}$ can be regarded as the low-pass filtered ageostrophic velocity. The high-pass and low-pass ageostrophic kinetic energy values are defined as: $E_{HP} = \mathbf{u_{HP}}^2/2$ and $E_{LP} = \mathbf{u_{LP}}^2/2$, respectively. The global data coverage and period of ageostrophic velocity and energy can be found in the Supplementary Note 1.

**Lagrangian chlorophyll variation rate.** By projecting the satellite observed log Chl to the location of surface drifter, the Lagrangian chlorophyll variation rate D(log Chl)/Dt can be directly computed along the drifter trajectory. Since the drifter data have higher temporal resolution (every 6 h) than the daily chlorophyll data, the log Chl data is first projected onto the drifter trajectory by linearly interpolating both temporally and spatially. Gaps in the observed snapshot of the chlorophyll map can be caused by satellite tracks positions and cloud coverage, a substantial portion of the drifter data points do not have observed chlorophyll values. The number of the effective drifter data points with valid log Chl value is 4,608,153 in total. The Lagrangian derivative D(log Chl)/Dt is computed along the drifter trajectory by linear fitting with a 1-day temporal window (five data points within each window). The one-day window size is selected according to the fastest phytoplankton growth rate determined by the cell division time-scale of ~1 day[57]. If a larger window size

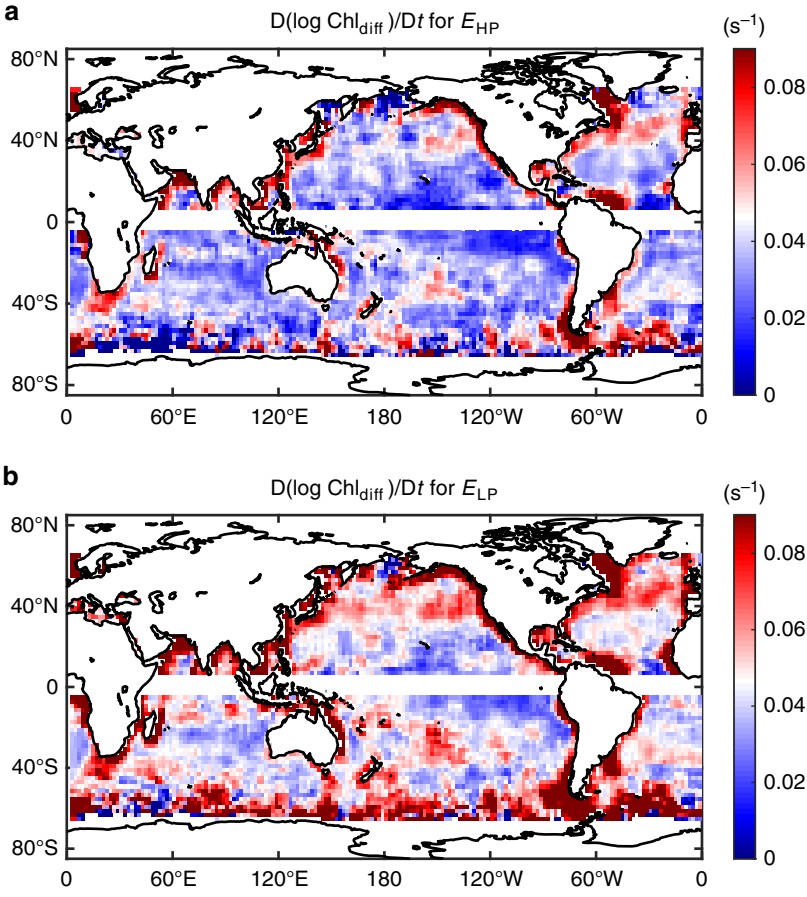

**Fig. 7** Maps of chlorophyll-increasing rate differences between energy levels. Difference of the chlorophyll-increasing rate D(log Chl$_\text{diff}$)/D$t$ computed according to **a** the 7-day high-pass ageostrophic kinetic energy $E_\text{HP}$ and **b** the 7-day low-pass ageostrophic kinetic energy $E_\text{LP}$. These properties are computed on a 2° × 2° global grid with a 3° × 3° moving average window. The difference of the chlorophyll-increasing rate of $E_\text{HP}$ is computed as D(log Chl$_\text{diff}$)/D$t$ = D(log Chl$_\text{strong}$)/D$t$ − D(log Chl$_\text{weak}$)/D$t$, where D(log Chl$_\text{strong}$)/D$t$ and D(log Chl$_\text{weak}$)/D$t$ are the averaged chlorophyll-increasing rates for $E_\text{HP} > 3*E_\text{HP0}$ and $E_\text{HP} < 1/3*E_\text{HP0}$ in the average window, respectively, and $E_\text{HP0}$ is the averaged $E_\text{HP}$ within the window. The difference of the chlorophyll-increasing rate of $E_\text{LP}$ is computed in the same way. Source data are provided as a Source Data file

is applied, the amplitude of the chlorophyll variation rate will be underestimated. The global data coverage and period of chlorophyll and its variation rate can be found in the Supplementary Note 1.

**Modified geostrophic vorticity**. The geostrophic vorticity $\zeta_g$ is computed by the surface geostrophic velocity anomaly $(u_g, v_g)$ from the altimeter data:

$$\zeta_g = \frac{\partial v_g}{\partial x} - \frac{\partial u_g}{\partial y}. \tag{4}$$

Considering cyclonic/anticyclonic eddies have $\zeta_g$ with different signs in the Northern/Southern hemispheres, we define a modified vorticity as $\omega = \zeta_g * \text{sign}(f)$, where $f$ is the Coriolis parameter. By this definition, cyclonic eddies always have positive modified vorticity $\omega > 0$ and anticyclonic eddies always have negative modified vorticity $\omega < 0$. The Lagrangian derivative of the modified vorticity D$\omega$/D$t$ is also computed along the drifter trajectory by linear fitting with a 1-day temporal window.

**Along-front coordinate and composition**. The strain saddle point is defined by a local strain rate maximum point with the geostrophic strain rate $S_{gc} > 1.0 \times 10^{-5}\,\text{s}^{-1}$ in the altimetry data. Along the trajectory of each surface drifter, geostrophic strain rate $S_g$, geostrophic velocity anomaly $(u_g, v_g)$, ageostrophic velocity $(u_a, v_a)$, ageostrophic kinetic energy $E_a$, surface temperature anomaly $T_a$, and Lagrangian chlorophyll variation rate D(log Chl)/D$t$ are readily accessible. The next step is to project the drifter data points onto an along-front coordinate defined as follows. For a given time, all strain saddle points at location $(x_s, y_s)$ are identified from altimetry maps. From these, only the points with the strain field stretching along the local geostrophic front are taken into account. In order to do this, the stretching direction is determined by the locally averaged angle $\theta_s$ with a 50-km radius averaging window. The along-front direction is determined by the locally averaged angle $\theta_g$ of

geostrophic current, also with a 50-km radius averaging window applied. The strain saddle points are finally selected if the criterion $\| \theta_s - \theta_g \| < 10°$ is satisfied.

Composition with regard to the strain saddle point is conducted as follows. Assume a simultaneous nearby drifter data point is located at $(x_d, y_d)$. The relative location of the drifter to the strain saddle point is $(x_c, y_c)$, where $x_c = (x_d - x_s)$ and $y_c = (y_d - y_s)$. The coordinate is rotated so that the along-front geostrophic velocity is pointed to the positive $x$-direction. In this case, the along-front coordinate of the drifter is given by $(x_r, y_r)$, where $x_r = x_c \cos(\theta_g) + y_c \sin(\theta_g)$ and $y_r = -x_c \sin(\theta_g) + y_c \cos(\theta_g)$. Notice that the geostrophic velocity anomaly $(u_g, v_g)$ and the ageostrophic velocity $(u_a, v_a)$ are also rotated accordingly. After the rotation, all properties are normalized by the center strain rate $S_{gc}$ to give their normalized values. Finally, the normalized properties are composited in the along-front coordinate $(x_r, y_r)$ to give their spatial structures around a mesoscale front with along-front strain stretching.

**Chlorophyll-increasing rate**. The chlorophyll-increasing rate is defined by the Lagrangian chlorophyll variation rate when the variation rate is positive D(log Chl)/D$t > 0$.

The observed chlorophyll-increasing rate is divided into different groups with strong or weak ageostrophic kinetic energy. For a given grid point in world ocean, all drifter data point within a 3-degree window is selected. The average high-pass/low-pass filtered ageostrophic kinetic energy within this window is readily computed as $E_\text{HP0}$ and $E_\text{LP0}$. The observed chlorophyll-increasing rate in divided into a strong energy group with $E_\text{HP} > 3*E_\text{HP0}$ (or $E_\text{LP} > 3*E_\text{LP0}$) and weak energy group with $E_\text{HP} < 1/3*E_\text{HP0}$ (or $E_\text{LP} < 1/3*E_\text{LP0}$). The averaged chlorophyll-increasing rates can be computed for both the strong/weak energy groups as D(log Chl$_\text{strong}$)/D$t$ and D(log Chl$_\text{weak}$)/D$t$. The difference between these two increasing rates is computed as D(log Chl$_\text{diff}$)/D$t$ = D(log Chl$_\text{strong}$)/D$t$ − D(log Chl$_\text{weak}$)/D$t$. This indicator is computed according to both high-pass/low-pass filtered ageostrophic kinetic energy at each grid point,

providing global maps of how chlorophyll-increasing rate changes with the high-pass/low-pass ageostrophic energy level.

## Data availability

The altimeter data can be accessed form website: https://www.aviso.altimetry.fr/en/data/products/sea-surface-height-products/global/gridded-sea-level-heights-and-derived-variables.html. The surface drifter dataset can be downloaded from the website: ftp://ftp.aoml.noaa.gov/phod/pub/buoydata. The chlorophyll data can be accessed from website: http://hermes.acri.fr/index.php?class=archive. The source data underlying Figs. 1 and 2, 4–7 and Supplementary Figs. 1–4 are provided as Source Data file.

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

## Acknowledgements

This research was supported by the National Programme on Global Change and Air-Sea Interaction under Grants GASI-IPOVAI-04 and the National Natural Science Foundation of China under Grants 41876001 and 41506007.

## Author contributions

Z.Z. and B.Q. contributed to design of the study, data processing and analysis, interpretation of the results, and writing of the manuscript. P.K. contributed to interpretation of the results and improving of the manuscript. S.T. contributed to data processing and improving of the manuscript.

## Additional information

**Competing interests:** The authors declare no competing interests.

