## [Peer Review File · Nature Communications]

Reviewers' comments:

Reviewer #1 (Remarks to the Author):

Review of the paper The Influence of Geostrophic Strain on Oceanic Ageostrophic Motion and Surface Chlorophyll by Zhang et al.

This is a very interesting, innovative and well-written paper that addresses one of the most important aspects of current days marine research, namely the interrelationships between geostrophic mezoscale and ageostrophic submesoscale components of the oceanic circulation, and their impact on the main oceanic primary producers – the phytoplankton.

It is now well acknowledged that submesoscale oceanic processes have an important role in upwelling nutrients and sustaining primary production in the upper well-lit surface layer of the ocean. Yet, despite their critical role, our understanding of the ecological, biogeochemical and climatic consequences of submesoscale processes is hindered by the difficulty to obtain relevant synoptic observations, largely due to their relatively short characteristic timescales and small characteristic spatial scales. The authors address this challenge by combining multi-satellite (altimetry and ocean color) and drifters data. The combined dataset is analyzed using an innovative Lagrangian method, whereby the satellite data are projected on drifter trajectories. This approach allows to untangle the fingerprint of internal (i.e. within a given water parcel) changes in satellite derived bio-optical properties from that of advection, in a way that is much more precise and robust than currently used Eulerian or Lagrangian analysis methods. Using this innovative methodology, which is likely to be used extensively in future research involving analysis of ocean color satellite data, the authors provide a unique global-scale perspective on the biogeochemical impact of submesoscale processes.

Based on the Lagrangian analysis of satellite and drifter data, the authors of this paper show strong relationships between surface concentration of chlorophyll – a proxy to phytoplankton biomass – local ageostrophic kinetic energy and geostrophic strain rate. This observed relationship reflects a situation in which strong strain rate enhances local ageostrophic kinetic energy that in turn favors the increase of near-surface chlorophyll. The paper also shows in a very clear way that the spatial structure of the strain-induced frontal processes is characterized by a cross-front ageostrophic secondary circulation with upwelling and chlorophyll increase along the light side of the density front. The results are very robust and unambiguous, providing the first observational evidence to the global effect of submesoscale dynamics on ocean productivity. These results shed new light on the interplay between ocean physics and biogeochemistry at the submesoscale, and improve substantially our ability to parameterize sub-grid processes in large scale carbon cycle and climate models.

In addition to its high scientific quality, the paper is written in a remarkably clear and easy-to follow way, making it highly accessible both for experts and for a wide readership with interest in the fields of oceanography, biogeochemistry, ecology, climate and remote sensing.

Below are a few minor comments and suggestions. Given its novelty, robustness, clarity and above all its contribution to our understanding of biophysical interaction in the marine environment, I strongly recommend accepting this manuscript for publication in Nature Communications.

Comments:

The map in Fig 1c shows that subtropical gyres are generally characterized by ageostrophic kinetic energy (E_a) levels substantially smaller than $0.1 \text{ m}^2 \text{ S}^{-2}$, which is the value above which $D(\log\text{Chl})/Dt$ becomes positive (Fig. 3a and lines 168-170 in the text). Does it imply that in these regions submesoscale ageostrophic processes are not likely to have significant impact on primary production?

Fig. 3a gives the impression that the large majority of the data points are associated with positive values of $D(\log\text{Chl})/Dt$. To my understanding this is not the case, and I recommend the authors refer to it in the text or in the figure caption.

In Fig. 1c ageostrophic kinetic energy (E_a) are shown for the range 0-0.06 $\text{m}^2 \text{S}^{-2}$, whereas in Fig. 3a it is shown for the range 0-1 $\text{m}^2 \text{S}^{-2}$. I recommend the authors refer to this difference in the text or in one of the figure captions, as it is a bit confusing.

If possible, it would be useful to have a rough estimate of vertical extension of the ageostrophic circulation. Importantly, having an idea on the depth from which the vertical motions originate may provide an indication on the source of the chlorophyll signature, as discussed in lines 292-297.

The authors should clarify how many pixels are averaged around the drifter location, when calculating the Lagrangian chlorophyll variation rate (lines 491-503).

Yoav Lehahn

Reviewer #2 (Remarks to the Author):

The results presented in this paper are certainly intriguing, but for several reasons, I am not convinced of the conclusions drawn. Firstly, the geostrophic surface velocities evaluated from satellite data have many issues and are not an accurate representation (even of the average within a period of time and space). Comparison with in-situ measurements show that there can be substantial errors in the position and magnitude of strong currents. The data from multiple altimeters (with a 10-day repeat cycle) is composited and gridded into a coarse resolution product (providing a nominal resolution of 0.25 - 1 deg, depending on how this is considered). The drifter data, on the other hand, gives the position of drifters from which the 15-m depth current can be calculated at the location of a drifter. Once again, compositing or binning goes into providing a gridded data set. Estimating an ageostrophic velocity from the difference of these two data sets, would result in errors that are too large and unconstrained to draw the conclusions that are presented in this paper. There is also no estimate or analysis of such errors.

The reason that the largest ageostrophic velocities are seen in the regions of the strongest currents, is likely because this is where the errors (and differences between the drifter and surface geostrophic velocities) are largest. The features with high geostrophic strain rates are at fronts, which is why the compositing shows certain patterns.

Re the chlorophyll, I am skeptical about whether the rate of change of chlorophyll can be calculated along drifter trajectories because satellite chlorophyll data has enormous gaps and the two data are not coincident. Hence evaluating the change of chlorophyll along a drifter trajectory is not reliable. If the data is composited or binned, then it has not adequately described in the Methods.

The paper starts discussing the results, without explaining what is being presented. The Methods section is not thorough and the compositing, averaging and binning of data is not well described. The time periods used for the evaluations are not described either.

Hence, even though the results are intriguing, I think that the patterns that are seen, are not confirmation of the processes claimed in the paper.

Minor points

Results - first para: The terms (and how they are calculated) should be described (even though details

can be given in the Methods). For example, the Results section starts discussing geostrophic kinetic energy and strain rate, without defining these or saying what they are based on. Similarly, the ageostrophic kinetic energy (Fig 1) is introduced without explanation. Fig 1. does not say what time period is used.

Also, what period are the ageostrophic velocities evaluated for, before they are squared for the kinetic energy?

Lines 140-143 — Is this shown using the data, or is it assumed to be true?

Lines 169-170 — Is this relationship between the ageostrophic kinetic energy and growth of chlorophyll expected for nutrient-limited regions?

Line 188-189 — not clear how the red line or grey shaded region is calculated

Line 282 — The figure says 2×10^{-5} deg C (which is a very small anomaly), where 2 deg is very large.

There are several other little errors or points that need clarification.

Reviewer #3 (Remarks to the Author):

This paper investigate the increase of surface chlorophyll induced by strain-induced frontal processes using an interesting combination of data sets: surface drifters, satellite altimetry and ocean-color. The main novelty is to consider here not the surface chlorophyll concentration but the Lagrangian chlorophyll variation along the drifter trajectories. The methodology that uses the global surface drifter data-set, provided by DAC, to estimate the ageostrophic velocity and the submesoscale kinetic energy was already applied in Zhang and Qui (2018). However, here the quantification of the Lagrangian chlorophyll variation open new perspective on the primary production in the euphotic layer. This manuscript, which provides a new methodology and highlights the role of local strain rate on surface chlorophyll could be of great interest for the oceanographic community. However, in the present stage, there is a lack of informations on the effective spatio-temporal coverage of the combined data-set. The authors focus on the correlations between the strain rate and the surface chlorophyll variations but other important dynamical variables or the seasonality of the submesoscale dynamics are not considered in this analysis.

Therefore, I invite the authors to revise their manuscript to address the specific concerns detailed in what follows.

1. What are the spatial and temporal coverages of the velocities derived from drifters trajectories ?

We guess, from the partial information given in the Methods section, that the global distribution of ageostrophic velocity components correspond to a mean value, averaged over a 18 years period (1993-2011) on $3^{\circ} \times 3^{\circ}$ boxes. However, the distribution of drifter data is not uniform worldwide. I guess some areas are much more sampled than others. In addition, the number of drifting buoys that were released almost doubled after 2005-2006. Hence, the informations on both the density of drifter measurements and their temporal distribution should be provided on a global map. The spatio-temporal distribution of these ageostrophic velocity components impacts the accuracy of the global maps of ageostrophic kinetic energy (Figure 1) and of the chlorophyll increasing rate (Figure 7).

2. Period of analysis of the chlorophyll variations rate and corresponding spatio-temporal coverages ?

In order to estimate the chlorophyll variations rate and compute the composite normalized properties in the along front-coordinates both the surface drifter velocities (1993-2011) and the ocean-color data-sets (1998-2017) should be combined. The temporal overlap of these two data sets lasts thirteen years (1998-2011). Here again, the spatio-temporal distribution of these combined observations is not uniform. Besides, due to the cloud coverage, these observations are less numerous (80% less) than for the geostrophic velocities only. Hence, here again the informations on both the density of these combined measurements (drifters+ ocean colors) and their temporal distribution should be provided on a global map.

3. What is the impact of horizontal divergence or vorticity variations on the chlorophyll variation rate ?

The figure 2 emphasizes the correlation between the Lagrangian chlorophyll variation rate and the geostrophic strain rate. However, correlation does not imply causation and other dynamical processes could also have a strong impact on the chlorophyll variation rate. The temporal variation of geostrophically balanced flows (formation or intensification of cyclonic eddies for instance) could also induce local and transient upwelling of the deep chlorophyll maxima towards the surface (Hasegawa et al. 2009; McGillicuddy, 2016). For these reasons the correlation of the Lagrangian chlorophyll variation rate with other dynamical variables such as the geostrophic divergence or the Lagrangian derivative of geostrophic vorticities should be added to the figure 2. In this perspective, the comparison between the temporal variations of positive and negative vorticities (e. g. cyclonic and anticyclonic eddies) could be very enlightening.

4. Is there any impact of seasonality on the relation between the geostrophic strain and the chlorophyll variation rate ?

The recent ultrahigh-resolution simulations of Su et al. (2018) clearly indicate that both submesoscale vorticity and the vertical transport exhibit a strong winter-peaked seasonality. Since the mixed layer is much deeper in winter the impact of the strain on mesoscale fronts will be more intense. Hence, the mechanism of strain-induced frontal process should be dominant in winter and probably less important in summer. A significant impact of seasonality on the relation between the geostrophic strain and the chlorophyll variation rate (figure 2) is expected. A simple comparison between winter and summer months could help to separate distinct processes.

Alex Stegner

Minor comments:

- "ageostrophic" lines 283 and 323

REFERENCES:

- Hasegawa, D., M. R. Lewis, and A. Gangopadhyay (2009), How islands cause phytoplankton to bloom in their wakes, *Geophys. Res. Lett.*, 36, L20605, doi:10.1029/2009GL039743.
- McGillicuddy D.J. Mechanisms of Physical- Biological-Biogeochemical Interaction at the Oceanic Mesoscale .*Annu. Rev. Mar. Sci.* 2016. 8:125–59.
- Su, Z., J. Wang , P. Klein, A. Thompson & D. Menemenlis Ocean submesoscales as a key component of the global heat budget, *Nature Comm.* (2018) 9:775, DOI: 10.1038/s41467-018-02983-w

Reviewer #1 (Remarks to the Author):

Review of the paper *The Influence of Geostrophic Strain on Oceanic Ageostrophic Motion and Surface Chlorophyll* by Zhang et al. This is a very interesting, innovative and well-written paper that addresses one of the most important aspects of current days marine research, namely the interrelationships between geostrophic mezoscale and ageostrophic submesoscale components of the oceanic circulation, and their impact on the main oceanic primary producers – the phytoplankton.

It is now well acknowledged that submesoscale oceanic process have an important role in upwelling nutrients and sustaining primary production in the upper well-lit surface layer of the ocean. Yet, despite their critical role, our understanding of the ecological, biogeochemical and climatic consequences of submesoscale processes is hindered by the difficulty to obtain relevant synoptic observations, largely due to their relatively short characteristic timescales and small characteristic spatial scales. The authors address this challenge by combining multi-satellite (altimetry and ocean color) and drifters data. The combined dataset is analyzed using an innovative Lagrangian method, whereby the satellite data are projected on drifter trajectories. This approach allows to untangle the fingerprint of internal (i.e. within a given water parcel) changes in satellite derived bio-optical properties from that of advection, in a way that is much more precise and robust than currently used Eulerian or Lagrangian analysis methods. Using this innovative methodology, which is likely to be used extensively in future researchers involving analysis of ocean color satellite data, the authors provide a unique global-scale perspective on the biogeochemical impact of submesoscale processes.

Based on the Lagrangian analysis of satellite and drifter data, the authors of this paper show strong relationships between surface concentration of chlorophyll – a proxy to phytoplankton biomass – local ageostrophic kinetic energy and geostrophic strain rate. This observed relationship reflects a situation in which strong strain rate enhances local ageostrophic kinetic energy that in turns favor the increase of near-surface chlorophyll. The paper also shows in a very clear way that the spatial structure of the strain-induced frontal processes is characterized by a cross-front ageostrophic secondary circulation with upwelling and chlorophyll increase along the light side of the density front. The results are very robust and unambiguous, providing the first observational evidence to the global effect of submesoscale dynamics on

ocean productivity. These results shed new light on the interplay between ocean physics and biogeochemistry at the submesoscale, and improve substantially our ability to parameterize sub-grid processes in large scale carbon cycle and climate models.

In addition to its high scientific quality, the paper is written in a remarkably clear and easy-to follow way, making it highly accessible both for experts and for a wide readership with interest in the fields of oceanography, biogeochemistry, ecology, climate and remote sensing. Below are a few minor comments and suggestions. Given its novelty, robustness, clarity and above all its contribution to our understanding of biophysical interaction in the marine environment, I strongly recommend accepting this manuscript for publication in Nature Communications.

We thank the reviewer for the positive assessment regarding the significance of our study and the novelty of our analysis methods. We are particularly pleased that the reviewer feels that our study would be of interest to the wider oceanographic community.

Comments:

The map in Fig 1c shows that subtropical gyres are generally characterized by ageostrophic kinetic energy (E_a) levels substantially smaller than $0.1 \text{ m}^2 \text{ S}^{-2}$, which is the value above which $D(\log\text{Chl})/Dt$ becomes positive (Fig. 3a and lines 168-170 in the text). Does it imply that in these regions submesoscale ageostrophic processes are not likely to have significant impact on primary production?

While Fig.1c shows that the climatological mean ageostrophic kinetic energy level is relatively small in regions like subtropical gyres, it does not preclude strong ageostrophic events (with ageostrophic kinetic energy E_a larger than $0.1 \text{ m}^2 \text{ s}^{-2}$) from happening within these regions. In Fig.R1 on next page, we compute the ratio between the number of data points with $E_a > 0.1 \text{ m}^2 \text{ s}^{-2}$ and the total number of data points within each average window. It shows that a substantial portion of the drifter data within strong current regions has E_a larger than $0.1 \text{ m}^2 \text{ s}^{-2}$ (nearly 40%-50%). At the same time, about 20%-30% of the drifter data in the western part of subtropical gyres has $E_a > 0.1 \text{ m}^2 \text{ s}^{-2}$. Since chlorophyll can bloom during these strong E_a events, we expect submesoscale ageostrophic processes will significantly contribute to the primary production even within the subtropical gyre.

Figure.R1 | Global distribution of the ratio between the number of data points with $E_a > 0.1 \text{ m}^2 \text{ s}^{-2}$ and the total number of data points within each $3^\circ \times 3^\circ$ average window.

Fig. 3a gives the impression that the large majority of the data points are associated with positive values of $D(\log\text{Chl})/Dt$. To my understanding this is not the case, and I recommend the authors refer to it in the text or in the figure caption.

A statement has been added to the manuscript in Line 177-184 (Please note all the Line Numbers here are according to the track-changed manuscript): “Although the ageostrophic kinetic energy E_a ranges from 0 to $1 \text{ m}^2 \text{ s}^{-2}$ in Fig.2a, about 18% data points are found to have $E_a > 0.1 \text{ m}^2 \text{ s}^{-2}$. This means only a small portion of the high ageostrophic events can effectively contribute to the chlorophyll increasing.”

In Fig. 1c ageostrophic kinetic energy (E_a) are shown for the range $0-0.06 \text{ m}^2 \text{ S}^{-2}$, whereas in Fig. 3a it is shown for the range $0-1 \text{ m}^2 \text{ S}^{-2}$. I recommend the authors refer to this difference in the text or in one of the figure captions, as it is a bit confusing.

Fig.1c is for the climatology mean value of E_a and Fig.2a shows the actual possible range for the E_a values. To avoid confusion, a statement has been added to the manuscript in Line 177-184: “Although the ageostrophic kinetic energy E_a is ranges from 0 to $1 \text{ m}^2 \text{ s}^{-2}$ in Fig.2a, about 18% data points are found to have $E_a > 0.1 \text{ m}^2 \text{ s}^{-2}$. This means only a small portion of the high ageostrophic events can effectively contribute to the chlorophyll increasing.”

If possible, it would be useful to have a rough estimate of vertical extension of the ageostrophic circulation. Importantly, having an idea on the depth from which the

vertical motions originate may provide an indication on the source of the chlorophyll signature, as discussed in lines 292-297.

According to recent theoretical considerations (Fox-Kemper et al., 2008; Klein and Lapeyre, 2009, Mahadevan, 2016; McWilliams, 2016) and numerical simulations (Capet et al., 2008; Fox-Kemper et al.; 2011), the vertical extension of the ageostrophic circulation is mostly confined to the local mixed layer depth. This is particularly true in winter when submesoscale motions are energetic because of the mixed-layer instabilities (Fox-Kemper et al., 2008). Note however some realistic simulations pointed to a vertical velocity field extending below the mixed-layer (Sasaki et al. 2014). This gives a very rough estimation of the vertical extension of the ageostrophic circulation about tens to two hundred meters in vertical direction, changing with the local stratification and rotation conditions (McWilliams, 2016). A statement has been added to the main text at Line 325-326.

The authors should clarify how many pixels are averaged around the drifter location, when calculating the Lagrangian chlorophyll variation rate (lines 491-503).

When using a one-day window to compute the chlorophyll variation rate, five points in each window are used to compute the variation rate by linear-fit. A statement has been added to the manuscript at Line 550.

Reviewer #2 (Remarks to the Author):

The results presented in this paper are certainly intriguing, but for several reasons, I am not convinced of the conclusions drawn. Firstly, the geostrophic surface velocities evaluated from satellite data have many issues and are not an accurate representation (even of the average within a period of time and space). Comparison with in-situ measurements show that there can be substantial errors in the position and magnitude of strong currents. The data from multiple altimeters (with a 10-day repeat cycle) is composited and gridded into a coarse resolution product (providing a nominal resolution of 0.25 - 1 deg, depending on how this is considered).

We are pleased that reviewer think our work is certainly intriguing and we thank the reviewer for bringing up several important points to help us improve our manuscript.

The altimetry data has been intensively used for investigating western boundary current and mesoscale eddies (both featured with strong surface current) for more than 20 years, and proved to be an effective tool (Imawaki et al., 2001; Fu et al. 2010; Morrow and Le Traon 2012; Chelton 2011b). The surface chlorophyll responses to mesoscale eddies have also been widely investigated by combining altimetry and ocean color remote sensing data (Chelton et al., 2011a; Gaube et al., 2013,2014). The recent review by Lehahn et al. (2018) clearly clarifies how mesoscale eddies have a structuring role in the phytoplankton distribution. The merging of multiple satellites has minimized the inhomogeneous sampling error, mapping error and interpolation error during the objective analysis processes. Furthermore, recent studies (see for example Ballarotta et al., 2019) have further emphasized that the currently used multi-satellite merged altimetry data can well reproduce the motions with wavelength of less than 100 km and can capture the eddies with a 40-50 km size at mid-latitude, or even ~ 25 km at high latitude. The error level of SSHA is on the order of one to several centimeters (Chelton 2011b). Considering that we mainly focus on strong ocean-front under large strain rate with typical SSHA perturbation about 10 to tens centimeters, the relative error here is about 10% - 20%.

The differ data, on the other hand, gives the position of drifters from which the 15-m depth current can be calculated at the location of a drifter. Once again, compositing or binning goes into providing a gridded data set. Estimating an ageostrophic velocity from the difference of these two data sets, would result in errors that are too large and unconstrained to draw the conclusions that are presented in this paper. There is also no estimate or analysis of such errors.

We used the original drifter data rather than a bin-average gridded data to compute the ageostrophic velocity. Although a formal estimation of errors in computing the ageostrophic velocity from drifter and altimetry data is not straightforward, our results detailed in Zhang and Qiu (2018) indicate that the ageostrophic signals are not substantially biased by observational error: Specifically, previous studies have shown that the observational errors tend to be larger when the local flow field is stronger (Chelton 2011b). This means that stronger eddies will have relatively higher level of observational errors. However, Zhang and Qiu (2018) show that the ageostrophic kinetic energy determined by the drifter data is weaker during eddy's mature phase

when the eddy has strongest amplitude. This can only be explained by the dynamical consequence of higher local strain field, and cannot be explained by the observational errors of altimetry and drifter data.

The reason that the largest ageostrophic velocities are seen in the regions of the strongest currents, is likely because this is where the errors (and differences between the drifter and surface geostrophic velocities) are largest. The features with high geostrophic strain rates are at fronts, which is why the compositing shows certain patterns.

In Fig.2f of Zhang and Qiu (2018), the positive relation between geostrophic strain rate and ageostrophic kinetic energy is confirmed when both are normalized by the local geostrophic kinetic energy. Since this normalization has ruled out the influence of the background strong currents, the positive relation between the geostrophic strain rate and ageostrophic kinetic energy is reliable and not caused by the observational errors of altimetry and drifter.

Re the chlorophyll, I am skeptical about whether the rate of change of chlorophyll can be calculated along drifter trajectories because satellite chlorophyll data has enormous gaps and the two data are not coincident. Hence evaluating the change of chlorophyll along a drifter trajectory is not reliable. If the data is composited or binned, then it has not adequately described in the Methods. The paper starts discussing the results, without explaining what is being presented. The Methods section is not thorough and the compositing, averaging and binning of data is not well described. The time periods used for the evaluations are not described either. Hence, even though the results are intriguing, I think that the patterns that are seen, are not confirmation of the processes claimed in the paper.

The chlorophyll data with gaps were not used in our study. We only selected the data when both the satellite chlorophyll and drifter data are simultaneously available. At the same time, our work mainly uses composite analysis to reveal the relation between the geostrophic strain, the ageostrophic energy level and the surface chlorophyll response. Without time series analysis involved, the data gap issue won't cause substantial errors to bias our main conclusions.

In connection to the data coverage issue, we have computed the global distributions of the number of data points of the ageostrophic kinetic energy E_a , the chlorophyll concentration and the chlorophyll variation rate. As shown in Fig.R2 below, there is no general consistency between the global patterns of the E_a data number distribution and the ageostrophic kinetic energy distribution in Fig.1c. There is also no general consistency between the global patterns of the distribution of the number of data points for chlorophyll and its variation rate $D(\log Chl)/Dt$ in Fig.7. These results imply that the global patterns of ageostrophic kinetic energy and chlorophyll variation rate are not biased by the data coverage. Statements have been added to the manuscript at Line 536-554 (Please note all the Line Numbers here are according to the track-changed manuscript).

Figure.R2 | Global distribution of the number of data points within each $3^\circ \times 3^\circ$ average window for (a) ageostrophic kinetic energy (b) chlorophyll concentration (c) chlorophyll variation rate.

We thank the reviewer for pointing out that the detailed information of the composite bin should be given: (1) In Fig.2a and Fig.2b, the chlorophyll variation rate is composited against the ageostrophic and geostrophic kinetic energy E_a and E_g , using a moving average window with a width $0.05 \text{ m}^2 \text{ s}^{-2}$. (2) In Fig.2c and Fig.2d, E_a and $D(\log Chl)/Dt$ are composited against the geostrophic strain rate S_g , using a moving average window with a width $0.1 \times 10^{-5} \text{ s}^{-1}$. (3) In Fig.4 and Fig.5, all parameters are composited in the rotated coordinate using a moving window with a bin size $25\text{km} \times 25\text{km}$. (4) In Fig.6b and Fig.6c, the chlorophyll variation rate is composited against the highpass and lowpass ageostrophic kinetic energy E_{HP} and E_{LP} , using a moving average window with a width $0.05 \text{ m}^2 \text{ s}^{-2}$. The information given above has been added to the corresponding figure captions.

Figure.R3 | The yearly variation of the total number of data points of (a) ageostrophic kinetic energy (b) chlorophyll concentration (c) chlorophyll variation rate.

We also thank the reviewer for pointing out the data period issue. Since there are three major data sets with different periods involved, more information should have been given in our original manuscript. When computing the ageostrophic velocity and kinetic energy, only the concurrent drifter and altimetry data are used. As shown in Fig.R3a, the period for ageostrophic velocity and kinetic energy is from 1993 to 2011. When computing the chlorophyll and its variation rate, only the concurrent drifter and ocean color data are used. As shown in Fig.R3b and Fig.R3c, the period for the chlorophyll and its variation rate is from 1998 to 2011. Since the period of the ageostrophic energy data covers that of the chlorophyll data, the period is from 1998 to 2011 whenever the calculation of the chlorophyll data is involved. Statements have been added to the main text at Line 536-554.

Minor points:

Results - first para: The terms (and how they are calculated) should be described (even though details can be given in the Methods). For example, the Results section starts discussing geostrophic kinetic energy and strain rate, without defining these or saying what they are based on. Similarly, the ageostrophic kinetic energy (Fig 1) is introduced without explanation. Fig 1. does not say what time period is used. Also, what period are the ageostrophic velocities evaluated for, before they are squared for the kinetic energy?

We thank the reviewer to point out this. Detailed definition and calculation method of this parameter are now available in the Method section. A statement referring to the Methods section is added to the main text at Line 125–151.

Lines 140-143 — Is this shown using the data, or is it assumed to be true?

This is a description of the physical processes related to the frontogenesis processes: the geostrophic strain field tends destroy the thermal wind balance, leading to ageostrophic motions that act to restore the thermal wind balance. This mechanism has been described in many papers (see reviews by Thomas et al. 2008; Klein and Lapeyre, 2009; Mahadavan 2016; McWilliams 2016). References have been added at Line 146 in the main text.

Lines 169-170 — Is this relationship between the ageostrophic kinetic energy and growth of chlorophyll expected for nutrient-limited regions?

Even in nutrient-limited regions, there exist vertical gradients of nutrients and chlorophyll concentration in the upper ocean layer because the maxima of nutrient and chlorophyll concentration are both located in subsurface. Thus, upwelling by ageostrophic motions will enhance the surface chlorophyll concentration and the positive relationship between the ageostrophic kinetic energy and growth of chlorophyll will be valid for the nutrient-limited regions. However, the intensity of the chlorophyll response to the local strain rate and ageostrophic motions in the nutrient-limited regions could be different from the global average value.

Line 188-189 — not clear how the red line or grey shaded region is calculated.

The red line represents the average value. In Fig.2a and Fig.2b, the chlorophyll variation rate is composited against the ageostrophic and geostrophic kinetic energy E_a and E_g , using a moving average window with a width $0.05 \text{ m}^2 \text{ s}^{-2}$. In Fig.2c and Fig.2d, E_a and $D(\log Chl)/Dt$ are composited against the geostrophic strain rate S_g , using a moving average window with a width $0.1 \times 10^{-5} \text{ s}^{-1}$. The grey shade represents the error range of averaging, computed by the standard error of the mean value as $Std/N^{1/2}$, where Std and N are the standard deviation and data number within each averaging bin, respectively. The above information has been added to the figure caption of Fig.2.

Line 282 — The figure says $2 \times 10^{-5} \text{ deg C}$ (which is a very small anomaly), where 2 deg is very large.

T_{an} in Fig.4c has a unit of $1 \times 10^5 \text{ deg C s}$. It represents the temperature anomaly normalized by the center strain rate S_{gc} . Considering that the typical in-situ temperature anomaly around an ocean front is about several degrees ($\sim 10^0 \text{ deg}$), and the center strain rate S_{gc} is about 10^{-5} s^{-1} , the amplitude of the normalized temperature anomaly $T_{an} = T_a/S_{gc}$ can be reasonably estimated to be 10^5 deg C s .

There are several other little errors or points that need clarification.

We thank the reviewer to remind us of this. We have gone through our manuscript carefully and corrected several minor errors, e.g. in Lines 394, 319, 351, 486, 572.

Reviewer #3 (Remarks to the Author):

This paper investigates the increase of surface chlorophyll induced by strain-induced frontal processes using an interesting combination of data sets: surface drifters, satellite altimetry and ocean-color. The main novelty is to consider here not the surface chlorophyll concentration but the Lagrangian chlorophyll variation along the drifter trajectories. The methodology that uses the global surface drifter data-set, provided by DAC, to estimate the ageostrophic velocity and the submesoscale kinetic energy was already applied in Zhang and Qiu (2018). However, here the quantification of the Lagrangian chlorophyll variation open new perspective on the primary production in the euphotic layer. This manuscript, which provides a new methodology and highlights the role of local strain rate on surface chlorophyll could be of great interest for the oceanographic community. However, in the present stage, there is a lack of information on the effective spatio-temporal coverage of the combined data-set. The authors focus on the correlations between the strain rate and the surface chlorophyll variations but other important dynamical variables or the seasonality of the submesoscale dynamics are not considered in this analysis.

Therefore, I invite the authors to revise their manuscript to address the specific concerns detailed in what follows.

We thank the reviewer for the positive assessment regarding the significance and novelty of our study and for bringing up the important data coverage issue to help us improve our manuscript.

1. What are the spatial and temporal coverages of the velocities derived from drifters trajectories?

We guess, from the partial information given in the Methods section, that the global distribution of ageostrophic velocity components correspond to a mean value, averaged over a 18 years period (1993-2011) on 3degx3deg boxes. However, the distribution of drifter data is not uniform worldwide. I guess some areas are much more sampled than others. In addition, the number of drifting buoys that were released almost doubled after 2005-2006. Hence, the information on both the density of drifter measurements and their temporal distribution should be provided on a global map. The spatio-temporal distribution of these ageostrophic velocity

components impacts the accuracy of the global maps of ageostrophic kinetic energy (Figure 1) and of the chlorophyll increasing rate (Figure 7).

We compute the global distributions of the number of data points of the ageostrophic kinetic energy E_a , the chlorophyll concentration and the chlorophyll variation rate. There is no general consistency between the global patterns of the of the distribution of the number of data points for E_a , as shown in Fig.R4a, and the ageostrophic kinetic energy distribution shown in Fig.1c. There is also no general consistency between the global patterns of the distribution of the number of data points for chlorophyll, as shown in in Fig.R4b-c and its variation rate $D(\log Chl)/Dt$ shown in Fig.7. Given this, we believe that the global patterns of ageostrophic kinetic energy and chlorophyll variation rate are not biased by the data coverage. Statements have been added to the main text at line 536-554 (Please note all the Line Numbers here are according to the track-changed manuscript).

Figure.R4 | Global distribution of the number of data points within each $3^\circ \times 3^\circ$ average window for (a) ageostrophic kinetic energy (b) chlorophyll concentration (c) chlorophyll variation rate.

We also thank the reviewer for pointing out the data period issue. Since there are three major data sets with different periods involved, more information should have been given in our original manuscript. When computing the ageostrophic velocity and kinetic energy, only the concurrent drifter and altimetry data are used. As shown in Fig.R3a, the period for ageostrophic velocity and kinetic energy is from 1993 to 2011. When computing the chlorophyll and its variation rate, only the concurrent drifter and ocean color data are used. As shown in Fig.R5b and Fig.R5c, the period for the chlorophyll and its variation rate is from 1998 to 2011. Since the period of the ageostrophic energy data covers that of the chlorophyll data, the period is from 1998 to 2011 whenever the calculation of the chlorophyll data is involved. Statements have been added to the main text at line 536-554.

Figure.R5 | The yearly variation of the total number of data points of (a) ageostrophic kinetic energy (b) chlorophyll concentration (c) chlorophyll variation rate.

2. Period of analysis of the chlorophyll variations rate and corresponding spatio-temporal coverages?

In order to estimate the chlorophyll variations rate and compute the composite normalized properties in the along front-coordinates both the surface drifter velocities (1993-2011) and the ocean-color data-sets (1998-2017) should be combined. The temporal overlap of these two data sets lasts thirteen years (1998-2011). Here again, the spatio-temporal distribution of these combined observations is not uniform. Besides, due to the cloud coverage, these observations are less numerous (80% less) than for the ageostrophic velocities only. Hence, here again the information on both the density of these combined measurements (drifters+ ocean colors) and their temporal distribution should be provided on a global map.

Please see our response to your specific concern 1.

3. What is the impact of horizontal divergence or vorticity variations on the chlorophyll variation rate?

The figure 2 emphasis the correlation between the Lagrangian chlorophyll variation rate and the geostrophic strain rate. However, correlation do not implies causation and other dynamical processes could also have a strong impact on the chlorophyll variation rate. The temporal variation of geostrophically balanced flows (formation or intensification of cyclonic eddies for instance) could also induce local and transient upwelling of the deep chlorophyll maxima towards the surface (Hasegawa et al. 2009; McGillicuddy, 2016). For these reasons the correlation of the Lagrangian chlorophyll variation rate with other dynamical variable such as the geostrophic divergence or the lagrangian derivative of geostrophic vorticities should be added to the figure 2. In this perspective, the comparison between the temporal variations of positive and negative vorticities (e. g. cyclonic and anticyclonic eddies) could be very enlightening.

We thank the reviewer for providing this insightful idea to test the chlorophyll variation rate from a different perspective. Based on the vorticity equation, the Lagrangian derivative of vorticity is related to the horizontal divergence. We can compute the geostrophic relative vorticity ζ by using altimetry data. In the northern hemisphere, when a cyclonic eddy enhances or an anticyclonic eddy decays, the Lagrangian derivative of vorticity will be positive, and an uplifting of isopycnal

surface is expected. In the opposite case, when an anticyclonic eddy enhances or a cyclonic eddy decays, the Lagrangian derivative of vorticity will be negative, and a downward motion of isopycnal surface is expected. In this sense, we expect upwelling will induce $D(\log_{10}Chl)/Dt > 0$ when $D\zeta/Dt > 0$. Conversely, when the Lagrangian derivative of vorticity $D\zeta/Dt < 0$, there will be downwelling and won't be a chlorophyll increase, or $D(\log_{10}Chl)/Dt \leq 0$. Notice that the sign of vorticity is opposite in the southern hemisphere, thus we define a modified vorticity as $\omega = \zeta * \text{sign}(f)$, where f is the Coriolis parameter. By this definition, $D\omega/Dt > 0$ is always related to a mesoscale upwelling in both northern and southern hemispheres.

Figure.R6 | Globally-averaged curves of chlorophyll variation rate $D(\log_{10}Chl)/Dt$ as a function of Lagrangian derivative of modified vorticity $D\omega/Dt$.

We compute the relation between the chlorophyll variation rate $D(\log_{10}Chl)/Dt$ and the Lagrangian derivative $D\omega/Dt$. As shown in Fig.R6, the result is just as expected: Increasing ω induces upwelling, and upwelling induces significant increase of chlorophyll; decreasing of ω induces downwelling, and downwelling results in no increase of chlorophyll. This result is consistent with the physical expectations by

Hasegawa et al. (2009) and McGillicuddy (2016) regarding the relationship between enhancing/weakening mesoscale eddies and upwelling /downwelling. In the revised manuscript, we have included the above results in the main text at Line 211-242 and in the Methods at Line 556-566. We have also added Fig.R6 as subfigure (e) in Fig.2 in the main text.

4. Is there any impact of seasonality on the relation between the geostrophic strain and the chlorophyll variation rate?

The recent ultrahigh-resolution simulations of Su et al. (2018) clearly indicates that both submesoscale vorticity and the vertical transport exhibit a strong winter-peaked seasonality. Since the mixed layer is much deeper in winter the impact of the strain on mesoscale fronts will be more intense. Hence, the mechanism of strain-induced frontal process should be dominant in winter a probably less important in summer. A significant impact of seasonality on the relation between the gesotrophic strain and the chlorophyll variation rate (figure 2) is expected. A simple comparison between winter and summer months could help to separate distinct processes.

Both the submesoscale energy level and the phytoplankton growth condition have strong seasonal cycle, seasonal variation of the chlorophyll response to the strain field is expected. Since the dataset is not large enough for a monthly computation of the relation between the geostrophic strain and the chlorophyll variation rate, we compute it only for the winter-spring half-year and summer-fall half-year as shown in Fig.R7.

The winter-spring half-year represents November-April for the northern hemisphere (May-October for the southern hemisphere). The summer-fall half-year represents May-October for the northern hemisphere (November-April for the southern hemisphere). We find that the chlorophyll response of the winter-spring half-year is stronger than the summer-fall half-year. This is consistent with the result that the submesoscale energy level is higher during the winter-spring half-year as pointed out by the reviewer. Since the seasonal circle is not the main focus of this paper, we leave its detailed investigation to a future study.

In the revised manuscript, we have added a statement in the main text at Line 206-210, and have included the above results in the Supplementary Information.

Figure.R7 | Globally-averaged curves of chlorophyll variation rate $D(\log Chl)/Dt$ as a function of local geostrophic strain rate S_g for the winter-spring half-year (blue curve) and summer-fall half-year (red curve). These functions are computed following the same procedure as Fig.2c.

Minor comments:

“ageostrophic” lines 283 and 323

We have corrected the words “ageostrophic” in these lines.

Reference:

Ballarotta, M. et al., On the resolution of ocean altimetry maps. *Ocean Sci. Discuss.*, <https://doi.org/10.5294/os-2018-156> (2019).

Capet, X., McWilliams, J. C., Molemaker, M. & Shchepetkin, A. Mesoscale to submesoscale transition in the California current system. Part 1: Flow structure, eddy flux and observational tests. *J. Phys. Oceanogr.* **38**, 29–43 (2008).

Chelton, D. B., Gaube, P., Schlax, M. G., Early, J. J. & Samelson, R. M. The influence of nonlinear mesoscale eddies on oceanic chlorophyll. *Science*, **334**, 328–332 (2011a).

- Chelton, D. B., Schlax, M. G. & Samelson, R. M. Global observations of nonlinear mesoscale eddies. *Prog. Oceanogr.*, **91**, 167–216, doi:10.1016/j.pocean.2011.01.002 (2011b).
- Fox-Kemper, B., Ferrari, R. & Hallberg, R. W. Parameterization of mixed layer eddies. I. Theory and diagnosis. *J. Phys. Ocean.* **38**, 1145–1165 (2008).
- Fu, L.-L., D. B. Chelton, P.-Y. Le Traon, and R. Morrow. Eddy dynamics from satellite altimetry. *Oceanography*, **23**, 14–25, doi:10.5670/oceanog.2010.02 (2010).
- Gaube, P., D. B. Chelton, P. G. Strutton & M. J. Behrenfeld. Satellite observations of chlorophyll, phytoplankton biomass and Ekman pumping in nonlinear mesoscale eddies. *J. Geophys. Res.* **118**, 6349–70 (2013).
- Gaube, P., D. J. McGillicuddy, D. B. Chelton, M. J. Behrenfeld & P. G. Strutton. Regional variations in the influence of mesoscale eddies on near-surface chlorophyll. *J. Geophys. Res. Oceans*. **119**, 8195–220 (2014).
- Hasegawa, D., Lewis, M. R. & Gangopadhyay, A. How islands cause phytoplankton to bloom in their wakes, *Geophys. Res. Lett.* **36**, L20605, doi:10.1029/2009GL039743 (2009).
- Imawaki, S. et al. Satellite altimeter monitoring the Kuroshio transport south of Japan. *Geophys. Res. Lett.* **28**, 17–20 (2001).
- Lapeyre, G. & Klein, P. Impact of the small-scale elongated filaments on the oceanic vertical pump. *J. Mar. Res.* **64**, 835–51 (2006).
- Lehahn, Y., d'Ovidio, F. & Koren, I. A satellite-based Lagrangian view on phytoplankton dynamics. *Annu. Rev. Mar. Sci.* **10**, 11.1-11.21 (2018).
- Klein, P. & Lapeyre, G. The oceanic vertical pump induced by mesoscale and submesoscale turbulence. *Annu. Rev. Mar. Sci.* **1**, 351–75 (2009).
- Mahadevan, A. The impact of submesoscale physics on primary productivity of plankton. *Annu. Rev. Mar. Sci.* **8**, 161–84 (2016).
- McGillicuddy D.J. Mechanisms of Physical- Biological-Biogeochemical Interaction at the Oceanic Mesoscale. *Annu. Rev. Mar. Sci.* **8**, 125–59 (2016).
- McWilliams, J. C., Colas, F. & Molemaker, M. Cold filamentary intensification and oceanic surface convergence lines. *Geophys. Res. Lett.* **36**, L18602 (2009).
- McWilliams, J. C. Submesoscale currents in the ocean. *Proc. R. Soc. A* **472**, 20160117 (2016).

- Morrow, R. & P. Le Traon. Recent advances in observing mesoscale ocean dynamics with satellite altimetry. *Advances in Space Research*. **50**, 1062-1076 (2012).
- Sasaki, H., Klein, P., Qiu, B. & Sasai, Y. Impact of oceanic-scale interactions on the seasonal modulation of ocean dynamics by the atmosphere. *Nat. Comm.* **5**, 5636. (2014).
- Thomas, L. N., Tandon, A. & Mahadevan, A. Submesoscale processes and dynamics, in *Eddy Resolving Ocean Modeling*, Geophys. Monogr. Ser., vol. 177, edited by M. W. Hecht and H. Hasumi, pp. 17–38, AGU, Washington, D. C., doi:10.1029/177GM04 (2008).
- Zhang, Z. & Qiu, B. Evolution of submesoscale ageostrophic motions through the life cycle of oceanic mesoscale eddies. *Geophys. Res. Lett.* **45**. <https://doi.org/10.1029/2018GL080399> (2018).

REVIEWERS' COMMENTS:

Reviewer #1 (Remarks to the Author):

The authors have thoroughly addressed my concerns, and I recommend accepting the manuscript for publication in Nature Communications. Yoav Lehahn

Following your request I have thoroughly went over the reviewers' comments and authors reply. To my understanding the authors have addressed the concerns raised by the reviewer in a satisfactory manner, which reinforces my view that the results and conclusions are based on an innovative and solid methodology.

Specifically, as noted correctly by the authors, there is now a large body of literature (some of which cited by the authors in their review) showing the usefulness of using satellite altimetry data for studying mesoscale ocean dynamics, and linking it to phytoplankton dynamics.

The authors clarify in their response that they used the original drifter data and not the gridded one (as this point, apparently, wasn't clear to begin with, the authors may consider emphasizing it in the paper). Furthermore, the approach taken by the authors for deriving ageostrophic velocities from drifter data relies on a very recent publication, which also deals with possible errors and biases.

Finally, since the authors only use data when both the satellite chlorophyll and drifter data are simultaneously available, the approach of combining the two datasets for extracting Lagrangian information on the chlorophyll field is actually very elegant and robust, and I don't see any difficulty in using it. This is actually a very elegant way to. While the sampling frequency along the drifter's trajectories is indeed reduced due to the gaps in the chlorophyll data reduces, it is reasonable to assume that it is sufficient for extracting meaningful Lagrangian information, as was done in this paper.

In summary, I find the methodology used in this work both innovative and correct, making it adequate for publication in Nature Communications.

Reviewer #1 (Remarks to the Author):

The authors have thoroughly addressed my concerns, and I recommend accepting the manuscript for publication in Nature Communications. Yoav Lehahn.

Following your request I have thoroughly went over the reviewers' comments and authors reply. To my understanding the authors have addressed the concerns raised by the reviewer in a satisfactory manner, which reinforces my view that the results and conclusions are based on an innovative and solid methodology.

Specifically, as noted correctly by the authors, there is now a large body of literature (some of which cited by the authors in their review) showing the usefulness of using satellite altimetry data for studying mesoscale ocean dynamics, and linking it to phytoplankton dynamics. The authors clarify in their response that they used the original drifter data and not the gridded one (as this point, apparently, wasn't clear to begin with, the authors may consider emphasizing it in the paper). Furthermore, the approach taken by the authors for deriving ageostrophic velocities from drifter data relies on a very recent publication, which also deals with possible errors and biases.

Finally, since the authors only use data when both the satellite chlorophyll and drifter data are simultaneously available, the approach of combining the two datasets for extracting Lagrangian information on the chlorophyll field is actually very elegant and robust, and I don't see any difficulty in using it. This is actually a very elegant way to. While the sampling frequency along the drifter's trajectories is indeed reduced due to the gaps in the chlorophyll data reduces, it is reasonable to assume that it is sufficient for extracting meaningful Lagrangian information, as was done in this paper.

In summary, I find the methodology used in this work both innovative and correct, making it adequate for publication in Nature Communications.

We thank the reviewer for the positive assessment regarding the novelty and reliability of our analysis methods. We are particularly pleased that the reviewer feels that our manuscript is adequate for publication in Nature Communications.

Following the reviewer's suggestion, a statement has been added to the revised manuscript in Line 167-169 to emphasize that the original drifter data is used in our study rather than the gridded data.